# SHAQ: Incorporating Shapley Value Theory into Multi-Agent Q-Learning

**Jianhong Wang**
Imperial College London, UK
jianhong.wang16@imperial.ac.uk

**Yuan Zhang**
University of Freiburg, Germany
yzhang@cs.uni-freiburg.de

**Yunjie Gu**[*]
University of Bath, UK
yg934@bath.ac.uk

**Tae-Kyun Kim**
KAIST, South Korea
kimtaekyun@kaist.ac.kr

## Abstract

Value factorisation is a useful technique for multi-agent reinforcement learning (MARL) in global reward game, however, its underlying mechanism is not yet fully understood. This paper studies a theoretical framework for value factorisation with interpretability via Shapley value theory. We generalise Shapley value to Markov convex game called *Markov Shapley value* (MSV) and apply it as a value factorisation method in global reward game, which is obtained by the equivalence between the two games. Based on the properties of MSV, we derive *Shapley-Bellman optimality equation* (SBOE) to evaluate the optimal MSV, which corresponds to an optimal joint deterministic policy. Furthermore, we propose *Shapley-Bellman operator* (SBO) that is proved to solve SBOE. With a stochastic approximation and some transformations, a new MARL algorithm called *Shapley Q-learning* (SHAQ) is established, the implementation of which is guided by the theoretical results of SBO and MSV. We also discuss the relationship between SHAQ and relevant value factorisation methods. In the experiments, SHAQ exhibits not only superior performances on all tasks but also the interpretability that agrees with the theoretical analysis. The implementation of this paper is placed on https://github.com/hsvgbkhgbv/shapley-q-learning.

## 1 Introduction

Cooperative games are a critical research area in multi-agent reinforcement learning (MARL). Many real-life tasks can be modeled as cooperative games, e.g. the coordination of autonomous vehicles [1], autonomous distributed logistics [2] and distributed voltage control in power networks [3]. In this paper, we consider global reward game (a.k.a. team reward game), an important subclass of cooperative games, wherein agents aim to jointly maximize cumulative global rewards over time. There are two categories of methods to solve this problem: (i) each agent identically maximizes cumulative global rewards, i.e. learning with a shared value function [4–6]; and (ii) each agent individually maximizes distributed values, i.e. learning with (implicit) credit assignments (e.g. marginal contribution and value factorisation) [7–11].

By the view of non-cooperative game theory, global reward game are equivalent to Markov game [12] with global reward (a.k.a. team reward). Its aim is to learn a stationary joint policy to reach a Markov equilibrium so that no agent tends to unilaterally change its policy to maximize cumulative global rewards. Standing by this view, learning with value factorisation cannot be directly explained [13]. In

---

[*]Correspondence to Yunjie Gu who is also an honorary lecturer at Imperial College London.

36th Conference on Neural Information Processing Systems (NeurIPS 2022).

this paper, to clearly interpret the value factorisation, we take the perspective of cooperative game theory [14], wherein agents are partitioned into coalitions and a payoff distribution scheme is found to distribute optimal values to coalitions. The corresponding solution is called Markov core, whereby no agent has an incentive to deviate. When all agents are partitioned into one coalition (called grand coalition), the payoff distribution scheme naturally plays the role of value factorisation.

Wang et al. [13] extended convex game (i.e. a game model in cooperative game theory) [14] to dynamic scenarios, which we name as Markov convex game in this paper. We construct the analytic form of Shapley value for Markov convex game, and prove that it reaches the Markov core under the grand coalition, named as Markov Shapley value. The optimal Markov Shapley value implies not only the optimal global value but also that no agent has incentives to deviate from the grand coalition. Additionally, Markov Shapley value enjoys the following properties: (i) identifiability of dummy agents; (ii) efficiency; (iii) reflecting the contribution; and (iv) symmetry. These properties aid the interpretation and validity of value factorisation in the global reward game, and such transparency and reliability are critical to industrial applications [3].

Based on the efficiency property, we derive Shapley-Bellman optimality equation that is an extension of Bellman optimality equation [15, 16]. Moreover, we propose Shapley-Bellman operator and prove its convergence to the Shapley-Bellman optimality equation and its optimal joint deterministic policy. With a stochastic approximation of Shapley-Bellman operator and some transformations, we derive an algorithm called Shapley Q-learning (SHAQ). SHAQ learns to approximate the optimal Markov Shapley Q-value (an equivalent form of the optimal Markov Shapley value). Moreover, we enable SHAQ decentralised in order to fit the decentralised execution framework and this decentralisation still remains the convergence condition of Shapley-Bellman operator.

The proposed method, SHAQ, is evaluated on two global reward games such as Predator-Prey [17] and multi-agent StarCraft benchmark tasks [18]. In the experiments, SHAQ shows not only generally good performances on solving all tasks but also the interpretability that is deficient in the state-of-the-art baselines.

## 2    Markov Convex Game

We now formally define Markov convex game (MCG) that can be described as a tuple $\langle \mathcal{N}, \mathcal{S}, \mathcal{A}, T, \Lambda, \pi, R_t, \gamma \rangle$. $\mathcal{N}$ is the set of all agents. $\mathcal{S}$ is the set of states and $\mathcal{A} = \times_{i \in \mathcal{N}} \mathcal{A}_i$ is the joint action set of all agents wherein $\mathcal{A}_i$ is each agent's action set. $T(\mathbf{s}, \mathbf{a}, \mathbf{s}') = Pr(\mathbf{s}'|\mathbf{s}, \mathbf{a})$ is defined as the transition probability between the successive states. $\mathcal{CS} = \{\mathcal{C}_1, ..., \mathcal{C}_n\}$ is a *coalition structure*, where $\mathcal{C}_i \subseteq \mathcal{N}$ called a *coalition* is a subset of all agents. $\Lambda$ is a collection of coalition structures. $\varnothing$ and $\mathcal{N}$ are two special cases of coalitions i.e. the *empty coalition* and the *grand coalition* respectively. Conventionally, it is assumed that $\mathcal{C}_m \bigcap \mathcal{C}_k = \varnothing, \forall \mathcal{C}_m, \mathcal{C}_k \subseteq \mathcal{N}$. $\pi = \times_{i \in \mathcal{N}} \pi_i$ is the joint policy of all agents. For any coalition $\mathcal{C}$, it is equipped with a *coalition policy* $\pi_{\mathcal{C}}(\mathbf{a}_{\mathcal{C}}|\mathbf{s}) = \times_{i \in \mathcal{C}} \pi_i(a_i|\mathbf{s})$ defined over the *coalition action set* $\mathcal{A}_{\mathcal{C}} = \times_{i \in \mathcal{C}} \mathcal{A}_i$. Therefore, $\pi$ can be seen as the *grand coalition policy*. $R_t : \mathcal{S} \times \mathcal{A}_{\mathcal{C}} \to [0, \infty)$ (i.e., a characteristic function) is the *coalition reward* at time step $t$. Accordingly, $R_t(\mathbf{s}, \mathbf{a})$ is the *grand coalition reward* (i.e., equivalent to the global reward) at time step $t$ that is written as $R(\mathbf{s}, \mathbf{a})$ or $R$ for conciseness in the rest of paper. $\gamma \in (0, 1)$ is the discounted factor. The infinite long-term discounted cumulative coalition rewards is defined as $V^{\pi_{\mathcal{C}}}(\mathbf{s}) = \mathbb{E}_{\pi_{\mathcal{C}}}\left[ \sum_{t=1}^{\infty} \gamma^{t-1} R_t(\mathbf{s}, \mathbf{a}_c) \mid \mathbf{S}_t = \mathbf{s} \right] \in [0, \infty)$, called a *coalition value*. Moreover, the empty coalition value $V^{\pi_\varnothing}(\mathbf{s}) = 0$ and $V^{\pi}(\mathbf{s})$ denotes the grand coalition value (i.e. also called the global value since the equivalence proof from [13]). The solution of MCG is to find a tuple $\langle \mathcal{CS}, (\max_{\pi_i} x_i(\mathbf{s}))_{i \in \mathcal{N}} \rangle$, where $(\max_{\pi_i} x_i(\mathbf{s}))_{i \in \mathcal{N}}$ indicates the *payoff distributions* (i.e. credit assignments) under the optimal joint policy given a coalition structure. Under the assumption $\mathcal{C}_m \bigcap \mathcal{C}_k = \varnothing, \forall \mathcal{C}_m, \mathcal{C}_k \subseteq \mathcal{N}$, the condition for MCG is as follows:

$$\max_{\pi_{\mathcal{C}_\cup}} V^{\pi_{\mathcal{C}_\cup}}(\mathbf{s}) \geq \max_{\pi_{\mathcal{C}_m}} V^{\pi_{\mathcal{C}_m}}(\mathbf{s}) + \max_{\pi_{\mathcal{C}_k}} V^{\pi_{\mathcal{C}_k}}(\mathbf{s}), \quad \forall \mathcal{C}_m, \mathcal{C}_k \subseteq \mathcal{N}, \mathcal{C}_\cup = \mathcal{C}_m \cup \mathcal{C}_k. \tag{1}$$

In MCG with the grand coalition i.e., $\mathcal{CS} = \{\mathcal{N}\}$, *Markov core*, a solution concept describing stability, is defined as a set of payoff distribution schemes by which no agent has incentives to deviate from the grand coalition to gain more profits. Mathematically, Markov core can be expressed as:

$$\texttt{MarkovCore} = \left\{ \left( \max_{\pi_i} x_i(\mathbf{s}) \right)_{i \in \mathcal{N}} \Big| \max_{\pi_{\mathcal{C}}} x(\mathbf{s}|\mathcal{C}) \geq \max_{\pi_{\mathcal{C}}} V^{\pi_{\mathcal{C}}}(\mathbf{s}), \forall \mathcal{C} \subseteq \mathcal{N}, \mathbf{s} \in \mathcal{S} \right\}, \tag{2}$$

where $\max_{\pi_\mathcal{C}} x(\mathbf{s}|\mathcal{C}) = \sum_{i \in \mathcal{C}} \max_{\pi_i} x_i(\mathbf{s})$. It aims to find a payoff distribution scheme $(x_i(\mathbf{s}))_{i \in \mathcal{N}}$ that can finally converge to Markov core under the optimal joint policy.

To assist the application on Q-learning, we similarly define *coalition Q-value* as $Q^{\pi_\mathcal{C}}(\mathbf{s}, \mathbf{a}_\mathcal{C}) \in [0, +\infty)$ for all coalitions $\mathcal{C} \subset \mathcal{N}$. Following the above convention, the grand coalition Q-value (or the global Q-value) can be written as $Q^\pi(\mathbf{s}, \mathbf{a})$. Moreover, the optimal coalition Q-value of $\mathcal{C}$ w.r.t. the optimal joint policy of $\mathcal{D} \subseteq \mathcal{C}$ (i.e., $\pi_\mathcal{D}^*$) and the suboptimal joint policy of $\mathcal{C} \backslash \mathcal{D}$ (i.e., $\pi_{\mathcal{C} \backslash \mathcal{D}}$) is defined as $Q^{\pi_\mathcal{D}^*}(\mathbf{s}, \mathbf{a}_\mathcal{C})$. Therefore, the optimal coalition Q-value of $\mathcal{C}$ w.r.t. the optimal joint policy of $\mathcal{C}$ is defined as $Q^{\pi_\mathcal{C}^*}(\mathbf{s}, \mathbf{a}_\mathcal{C})$. Accordingly, the optimal global coalition Q-value w.r.t. the optimal joint policy of the grand coalition is denoted as $Q^{\pi^*}(\mathbf{s}, \mathbf{a})$.

# 3 Markov Shapley Value

By the view of cooperative game theory, the grand coalition is progressively formed by a permutation of agents. Accordingly, marginal contribution is an implementation of the credit reflecting an agent's contribution. The formal definition is shown in Definition 1.

**Definition 1.** *In Markov convex game, with a permutation of agents $\langle j_1, j_2, ..., j_{|\mathcal{N}|} \rangle$, $\forall j_n \in \mathcal{N}$ forming the grand coalition $\mathcal{N}$, where $n \in \{1, ..., |\mathcal{N}|\}$, $j_a \neq j_b$ if $a \neq b$, the marginal contribution of an agent $i$ is defined as the following equation such that*

$$\Phi_i(\mathbf{s}|\mathcal{C}_i) = \max_{\pi_{\mathcal{C}_i}} V^{\pi_{\mathcal{C}_i \cup \{i\}}}(\mathbf{s}) - \max_{\pi_{\mathcal{C}_i}} V^{\pi_{\mathcal{C}_i}}(\mathbf{s}), \tag{3}$$

*where $\mathcal{C}_i = \{j_1, ..., j_{n-1}\}$ for $j_n = i$ is an arbitrary intermediate coalition where agent $i$ would join during the process of grand coalition formation.*

**Proposition 1.** *Agent $i$'s action marginal contribution can be derived as follows:*

$$\Phi_i(\mathbf{s}, a_i|\mathcal{C}_i) = \max_{\mathbf{a}_{\mathcal{C}_i}} Q^{\pi_{\mathcal{C}_i}^*}(\mathbf{s}, \mathbf{a}_{\mathcal{C}_i \cup \{i\}}) - \max_{\mathbf{a}_{\mathcal{C}_i}} Q^{\pi_{\mathcal{C}_i}^*}(\mathbf{s}, \mathbf{a}_{\mathcal{C}_i}). \tag{4}$$

As Proposition 1 shows, an agent's action marginal contribution (analogous to Q-value) can be derived according to Eq.4. It is usually more useful for solving MARL problems.

It is apparent that marginal contribution only considers one permutation to form the grand coalition. By the viewpoint from Shapley [19], the fairness is achieved through considering how much the agent $i$ increases the optimal values (i.e. marginal contributions) of the coalitions in all possible permutations when it joins in, i.e., $\max_{\pi_{\mathcal{C}_i}} V^{\pi_{\mathcal{C}_i \cup \{i\}}}(\mathbf{s}) - \max_{\pi_{\mathcal{C}_i}} V^{\pi_{\mathcal{C}_i}}(\mathbf{s})$, $\forall \mathcal{C}_i \subseteq \mathcal{N} \backslash \{i\}$. Therefore, we construct Shapley value under Markov dynamics based on the marginal contributions shown in Definition 2, named as *Markov Shapley value* (MSV).

**Definition 2.** *Markov Shapley value is represented as*

$$V_i^\phi(\mathbf{s}) = \sum_{\mathcal{C}_i \subseteq \mathcal{N} \backslash \{i\}} \frac{|\mathcal{C}_i|!(|\mathcal{N}| - |\mathcal{C}_i| - 1)!}{|\mathcal{N}|!} \cdot \Phi_i(\mathbf{s}|\mathcal{C}_i). \tag{5}$$

*With the deterministic policy, Markov Shapley value can be equivalently represented as*

$$Q_i^\phi(\mathbf{s}, a_i) = \sum_{\mathcal{C}_i \subseteq \mathcal{N} \backslash \{i\}} \frac{|\mathcal{C}_i|!(|\mathcal{N}| - |\mathcal{C}_i| - 1)!}{|\mathcal{N}|!} \cdot \Phi_i(\mathbf{s}, a_i|\mathcal{C}_i). \tag{6}$$

*where $\Phi_i(\mathbf{s}|\mathcal{C}_i)$ is defined in Eq.3 and $\Phi_i(\mathbf{s}, a_i|\mathcal{C}_i)$ is defined in Eq.4.*

For convenience, we name Eq.6 as *Markov Shapley Q-value* (MSQ). Briefly, MSV calculates the weighted average of marginal contributions. Since a coalition may repeatedly appear among all permutations (i.e. $|\mathcal{N}|!$ permutations), the ratio between the occurrence frequency $|\mathcal{C}_i|!(|\mathcal{N}| - |\mathcal{C}_i| - 1)!$ and the total frequency $|\mathcal{N}|!$ is used as a weight to describe the importance of the corresponding marginal contribution. Besides, the sum of all weights is equal to 1, so each weight can be interpreted as a probability distribution. Consequently, MSV can be seen as the expectation of marginal contributions, denoted as $\mathbb{E}_{\mathcal{C}_i \sim Pr(\mathcal{C}_i|\mathcal{N} \backslash \{i\})}[\Phi_i(\mathbf{s}|\mathcal{C}_i)]$. Note that $Pr(\mathcal{C}_i|\mathcal{N} \backslash \{i\})$ is a bell-shaped probability distribution. By the above relationship, Remark 1 is directly obtained.

**Remark 1.** *Uniformly sampling different permutations is equivalent to directly sampling from $Pr(\mathcal{C}_i|\mathcal{N} \backslash \{i\})$, since the coalition generation is from the permutation to form the grand coalition.*

**Proposition 2.** *Markov Shapley value possesses properties as follows: (i) identifiability of dummy agents:* $V_i^\phi(\mathbf{s}) = 0$; *(ii) efficiency:* $\max_\pi V^\pi(\mathbf{s}) = \sum_{i \in \mathcal{N}} \max_{\pi_i} V_i^\phi(\mathbf{s})$; *(iii) reflecting the contribution; and (iv) symmetry.*

Proposition 2 shows four properties of MSV. The most important property is Property (ii) that aids the formulation of Shapley-Bellman optimality equation. Property (iii) shows that MSV is a fundamental index to quantitatively describe each agent's contribution. Property (i) and (iii) play important roles in interpretation for value factorisation (or credit assignment). Property (iv) indicates that if two agents are symmetric, then their optimal MSVs should be equal, *but the reverse does not necessarily hold*. All these properties that define the fairness are inherited from the original Shapley value [19].

## 4 Shapley Q-Learning

### 4.1 Definition and Formulation

**Shapley-Bellman Optimality Equation.** Based on the Bellman optimality equation [15] and the following conditions (the interpretability of which are left to Section 4.2):

**C.1.** Efficiency of MSV (i.e. the result from Proposition 2);

**C.2.** $Q_i^{\phi^*}(\mathbf{s}, a_i) = w_i(\mathbf{s}, a_i) \, Q^{\pi^*}(\mathbf{s}, \mathbf{a}) - b_i(\mathbf{s})$, where $w_i(\mathbf{s}, a_i) > 0$ and $b_i(\mathbf{s}) \geq 0$ are bounded and $\sum_{i \in \mathcal{N}} w_i(\mathbf{s}, a_i)^{-1} b_i(\mathbf{s}) = 0$,

we derive *Shapley-Bellman optimality equation* (SBOE) for evaluating the optimal MSQ (an equivalent form to optimal MSV) such that

$$\mathbf{Q}^{\phi^*}(\mathbf{s}, \mathbf{a}) = \mathbf{w}(\mathbf{s}, \mathbf{a}) \sum_{\mathbf{s}' \in \mathcal{S}} Pr(\mathbf{s}'|\mathbf{s}, \mathbf{a}) \Big[ R + \gamma \sum_{i \in \mathcal{N}} \max_{a_i} Q_i^{\phi^*}(\mathbf{s}', a_i) \Big] - \mathbf{b}(\mathbf{s}), \tag{7}$$

where $\mathbf{w}(\mathbf{s}, \mathbf{a}) = [w_i(\mathbf{s}, a_i)]^\top \in \mathbb{R}_+^{|\mathcal{N}|}$; $\mathbf{b}(\mathbf{s}) = [b_i(\mathbf{s})]^\top \in \mathbb{R}_{\geq 0}^{|\mathcal{N}|}$; $\mathbf{Q}^{\phi^*}(\mathbf{s}, \mathbf{a}) = [Q_i^{\phi^*}(\mathbf{s}, a_i)]^\top \in \mathbb{R}_{\geq 0}^{|\mathcal{N}|}$ and $Q_i^{\phi^*}(\mathbf{s}, a_i)$ denotes the optimal MSQ. If Eq.7 holds, the optimal MSQ is achieved. Moreover, it reveals an implication that for any $\mathbf{s} \in \mathcal{S}$ and $a_i^* = \arg\max_{a_i} Q_i^{\phi^*}(\mathbf{s}, a_i)$, we have a solution $w_i(\mathbf{s}, a_i^*) = 1/|\mathcal{N}|$ (see Appendix E.4.1). Literally, the assigned credits would be equal and each agent would receive $Q^{\pi^*}(\mathbf{s}, \mathbf{a})/|\mathcal{N}|$ if performing the optimal actions. It is apparent that the efficiency still holds under this situation, which can be interpreted as an extremely fair credit assignment such that the credit to each agent should not be discriminated if all of them perform optimally, regardless of their roles. The equal credit assignment was also revealed by Wang et al. [20] recently from another perspective of analysis. Nevertheless, $w_i(\mathbf{s}, a_i)$ for $a_i \neq \arg\max_{a_i} Q_i^{\phi^*}(\mathbf{s}, a_i)$ needs to be learned.

**Shapley-Bellman Operator.** To find an optimal solution described by Eq.7, we now propose an operator called *Shapley-Bellman operator* (SBO), i.e., $\Upsilon : \times_{i \in \mathcal{N}} Q_i^\phi(\mathbf{s}, a_i) \mapsto \times_{i \in \mathcal{N}} Q_i^\phi(\mathbf{s}, a_i)$, which is defined as follows:

$$\Upsilon \Big( \times_{i \in \mathcal{N}} Q_i^\phi(\mathbf{s}, a_i) \Big) = \mathbf{w}(\mathbf{s}, \mathbf{a}) \sum_{\mathbf{s}' \in \mathcal{S}} Pr(\mathbf{s}'|\mathbf{s}, \mathbf{a}) \Big[ R + \gamma \sum_{i \in \mathcal{N}} \max_{a_i} Q_i^\phi(\mathbf{s}', a_i) \Big] - \mathbf{b}(\mathbf{s}), \tag{8}$$

where $w_i(\mathbf{s}, a_i) = 1/|\mathcal{N}|$ when $a_i = \arg\max_{a_i} Q_i^\phi(\mathbf{s}, a_i)$. We prove that the optimal joint deterministic policy can be achieved by recursively running SBO in Theorem 1.

**Theorem 1.** *Shapley-Bellman operator is able to converge to the optimal Markov Shapley Q-value and the corresponding optimal joint deterministic policy when* $\max_{\mathbf{s}} \big\{ \sum_{i \in \mathcal{N}} \max_{a_i} w_i(\mathbf{s}, a_i) \big\} < \frac{1}{\gamma}$.

**Shapley Q-Learning.** For easy implementation, we conduct transformation for the stochastic approximation of SBO and derive *Shapley Q-learning* (SHAQ) whose TD error is shown as follows:

$$\Delta(\mathbf{s}, \mathbf{a}, \mathbf{s}') = R + \gamma \sum_{i \in \mathcal{N}} \max_{a_i} Q_i^\phi(\mathbf{s}', a_i) - \sum_{i \in \mathcal{N}} \delta_i(\mathbf{s}, a_i) \, Q_i^\phi(\mathbf{s}, a_i), \tag{9}$$

where

$$\delta_i(\mathbf{s}, a_i) = \begin{cases} 1 & a_i = \arg\max_{a_i} Q_i^\phi(\mathbf{s}, a_i), \\ \alpha_i(\mathbf{s}, a_i) & a_i \neq \arg\max_{a_i} Q_i^\phi(\mathbf{s}, a_i). \end{cases} \tag{10}$$

Actually, the closed-form expression of $\delta_i(\mathbf{s}, a_i)$ is written as $|\mathcal{N}|^{-1} w_i(\mathbf{s}, a_i)^{-1}$. If inserting the condition that $w_i(\mathbf{s}, a_i) = 1/|\mathcal{N}|$ when $a_i = \arg\max_{a_i} Q_i^\phi(\mathbf{s}, a_i)$ as well as defining $\delta_i(\mathbf{s}, a_i)$ as $\alpha_i(\mathbf{s}, a_i)$ when $a_i \neq \arg\max_{a_i} Q_i^\phi(\mathbf{s}, a_i)$, Eq. 10 is obtained. The term $\mathbf{b}(\mathbf{s})$ is cancelled in Eq. 9 thanks to the condition such that $\sum_{i \in \mathcal{N}} w_i(\mathbf{s}, a_i)^{-1} b_i(\mathbf{s}) = 0$. Note that the condition to $w_i(\mathbf{s}, a_i)$ in Theorem 1 should hold for the convergence of SHAQ in implementation (see Appendix E.4.4).

## 4.2 Validity and Interpretability

In this section, we show the validity of SBOE and the interpretability of SHAQ, i.e., providing the reasons why SBOE is valid to be formulated and SHAQ is an interpretable value factorisation method for the global reward game.

**Theorem 2.** *The optimal Markov Shapley value is a solution in the Markov core under Markov convex game with the grand coalition.*

**Remark 2.** *For an arbitrary state $\mathbf{s} \in \mathcal{S}$, by C.2 it is not difficult to check that even if an arbitrary agent $i$ is dummy (i.e., $Q_i^{\phi^*}(\mathbf{s}, a_i) = 0$ for some $i \in \mathcal{N}$), $Q^{\pi^*}(\mathbf{s}, \mathbf{a})$ and $Q_j^{\phi^*}(\mathbf{s}, a_j), \forall j \neq i$ would not be zero if $b_i(\mathbf{s}) \neq 0$. If the extreme case happens that for an arbitrary state $\mathbf{s} \in \mathcal{S}$ all agents are dummies, since $\sum_{i \in \mathcal{N}} w_i(\mathbf{s}, a_i)^{-1} b_i(\mathbf{s}) = 0$ we are allowed to set $b_i(\mathbf{s}) = 0, \forall i \in \mathcal{N}$ so that $Q^{\pi^*}(\mathbf{s}, \mathbf{a}) = 0$ and efficiency such that $\max_{\mathbf{a}} Q^{\pi^*}(\mathbf{s}, \mathbf{a}) = \sum_{i \in \mathcal{N}} \max_{a_i} Q_i^{\phi^*}(\mathbf{s}, a_i)$ is still valid.*

First, we give a proof for showing that the optimal MSV is a solution in Markov core under the grand coalition, as Theorem 2 shows. Since a solution in Markov core implies the optimal global value (see Remark 5 in Appendix D.2.2), we can conclude that *the optimal MSV can lead to the optimal global value* (a.k.a. social welfare), which links Condition C.1 to Markov core. As a result, *solving SBOE is equivalent to solving Markov core under the grand coalition and SHAQ is actually a learning algorithm that reliably converges to Markov core*. As per the definition in Section 2, we can say that SHAQ leads to the result that no agents have incentives to deviate from the grand coalition, which provides an interpretation of value factorisation for global reward game. Condition C.2 is a condition that *maintains the validity of the relationship between the optimal MSQ and the optimal global Q-value even if there exist dummy agents* (see Remark 2), so that the definition of SBOE is valid for MCG and MSQ in almost every case, which preserves the completeness of the theory.

## 4.3 Implementations

We now describe a practical implementation of SHAQ for Dec-POMDP [21] (i.e. the global reward game but with partial observations). First, the global state is replaced by the history of each agent to guarantee the optimal deterministic joint policy [21]. Accordingly, Markov Shapley Q-value is denoted as $Q_i^\phi(\tau_i, a_i)$, wherein $\tau_i$ is a history of partial observations of agent $i$. Since the paradigm of centralised training decentralised execution (CTDE) [22] is applied, the global state (i.e. $\mathbf{s}$) for $\hat{\alpha}_i(\mathbf{s}, a_i)$ can be obtained during training.

**Proposition 3.** *Suppose any action marginal contribution can be factorised to the form such that $\Phi_i(\mathbf{s}, a_i | \mathcal{C}_i) = \sigma(\mathbf{s}, \mathbf{a}_{\mathcal{C}_i \cup \{i\}}) \hat{Q}_i(\mathbf{s}, a_i)$. With the condition such that*

$$\mathbb{E}_{\mathcal{C}_i \sim Pr(\mathcal{C}_i | \mathcal{N} \setminus \{i\})} \left[ \sigma(\mathbf{s}, \mathbf{a}_{\mathcal{C}_i \cup \{i\}}) \right] = \begin{cases} 1 & a_i = \arg\max_{a_i} Q_i^\phi(\mathbf{s}, a_i), \\ K \in (0, 1) & a_i \neq \arg\max_{a_i} Q_i^\phi(\mathbf{s}, a_i), \end{cases}$$

*we have*

$$\begin{cases} Q_i^\phi(\mathbf{s}, a_i) = \hat{Q}_i(\mathbf{s}, a_i) & a_i = \arg\max_{a_i} \hat{Q}_i(\mathbf{s}, a_i), \\ \alpha_i(\mathbf{s}, a_i) Q_i^\phi(\mathbf{s}, a_i) = \hat{\alpha}_i(\mathbf{s}, a_i) \hat{Q}_i(\mathbf{s}, a_i) & a_i \neq \arg\max_{a_i} \hat{Q}_i(\mathbf{s}, a_i), \end{cases} \tag{11}$$

*where $\hat{\alpha}_i(\mathbf{s}, a_i) = \mathbb{E}_{\mathcal{C}_i \sim Pr(\mathcal{C}_i | \mathcal{N} \setminus \{i\})} \left[ \hat{\psi}_i(\mathbf{s}, a_i; \mathbf{a}_{\mathcal{C}_i}) \right]$ and $\hat{\psi}_i(\mathbf{s}, a_i; \mathbf{a}_{\mathcal{C}_i}) := \alpha_i(\mathbf{s}, a_i) \sigma(\mathbf{s}, \mathbf{a}_{\mathcal{C}_i \cup \{i\}})$.*

Compatible with the decentralised execution, we use only one parametric function $\hat{Q}_i(\tau_i, a_i)$ to directly approximate $Q_i^\phi(\tau_i, a_i)$. By inserting Eq. 11 into Eq. 9, $\delta_i(\mathbf{s}, a_i)$ is transformed into the form as follows:

$$\hat{\delta}_i(\mathbf{s}, a_i) = \begin{cases} 1 & a_i = \arg\max_{a_i} \hat{Q}_i(\mathbf{s}, a_i), \\ \hat{\alpha}_i(\mathbf{s}, a_i) & a_i \neq \arg\max_{a_i} \hat{Q}_i(\mathbf{s}, a_i), \end{cases} \tag{12}$$

where $\hat{\alpha}_i(\mathbf{s}, a_i) = \mathbb{E}_{\mathcal{C}_i \sim Pr(\mathcal{C}_i | \mathcal{N} \setminus \{i\})} \left[ \hat{\psi}_i(\mathbf{s}, a_i; \mathbf{a}_{\mathcal{C}_i}) \right]$. To solve partial observability, $\hat{Q}_i(\tau_i, a_i)$ is empirically represented as recurrent neural network (RNN) with GRUs [23]. $\hat{\psi}_i(\mathbf{s}, a_i; \mathbf{a}_{\mathcal{C}_i})$ is directly approximated by a parametric function $F_{\mathbf{s}} + 1$ and thus $\hat{\alpha}_i(\mathbf{s}, a_i)$ can be expressed as follows:

$$\hat{\alpha}_i(\mathbf{s}, a_i) = \frac{1}{M} \sum_{k=1}^{M} F_{\mathbf{s}} \left( \hat{Q}_{\mathcal{C}_i^k}(\tau_{\mathcal{C}_i^k}, \mathbf{a}_{\mathcal{C}_i^k}), \hat{Q}_i(\tau_i, a_i) \right) + 1, \tag{13}$$

where $\hat{Q}_{\mathcal{C}_i^k}(\tau_{\mathcal{C}_i^k}, \mathbf{a}_{\mathcal{C}_i^k}) = \frac{1}{|\mathcal{C}_i^k|} \sum_{j \in \mathcal{C}_i^k} \hat{Q}_j(\tau_j, a_j)$ and $\mathcal{C}_i^k$ is sampled $M$ times from $Pr(\mathcal{C}_i | \mathcal{N} \setminus \{i\})$ (i.e., implemented as Remark 1 suggests) to approximate $\mathbb{E}_{\mathcal{C}_i \sim Pr(\mathcal{C}_i | \mathcal{N} \setminus \{i\})} [\hat{\psi}_i(\mathbf{s}, a_i; \mathbf{a}_{\mathcal{C}_i})]$ using Monte Carlo approximation; and $F_{\mathbf{s}}$ is a monotonic function, followed by an absolute activation function, whose weights are generated from hyper-networks w.r.t. the global state. We show that Eq.13 satisfies the condition to $w_i(\mathbf{s}, a_i)$ in Theorem 1 (see Appendix E.6.1), so it is a reliable implementation.

By using the framework of fitted Q-learning [24] to solve large number of states (i.e., could be usually infinite) and plugging in the above designed modules, the practical least-square-error loss function derived from Eq.9 is therefore stated as follows:

$$\min_{\theta, \lambda} \mathbb{E}_{\mathbf{s}, \tau, \mathbf{a}, R, \tau'} \left[ \left( R + \gamma \sum_{i \in \mathcal{N}} \max_{a_i} \hat{Q}_i(\tau_i', a_i; \theta^-) - \sum_{i \in \mathcal{N}} \hat{\delta}_i(\mathbf{s}, a_i; \lambda) \hat{Q}_i(\tau_i, a_i; \theta) \right)^2 \right], \tag{14}$$

where all agents share the parameters of $\hat{Q}_i(\mathbf{s}, a_i; \theta)$ and $\hat{\alpha}_i(\mathbf{s}, a_i; \lambda)$ respectively; and $\hat{Q}_i(\mathbf{s}', a_i; \theta^-)$ works as the target where $\theta^-$ is periodically updated. The general training procedure follows the paradigm of DQN [25], with a replay buffer to store the online collection of agents' episodes. To depict an overview of the algorithm, the pseudo code is shown in Appendix A.

## 5 Related Work

**Value Factorisation in MARL.** To deal with the instability during training in global reward game by independent learners [26], the centralised training and decentralised execution (CTDE) [22] was proposed and it became a general paradigm for MARL. Based on CTDE, MADDPG [27] learns a global Q-value that can be regarded as assigning the same credits to all agents during training [13], which may cause the unfair credit assignment [28]. To avoid this problem, VDN [8] was proposed to learn the factorised Q-value, assuming that any global Q-value equals to the sum of decentralised Q-values. Nevertheless, this factorisation may limit the representation of the global Q-value. To mitigate this issue, QMIX [9] and QTRAN [10] were proposed to represent the global Q-value with a richer class w.r.t. decentralised Q-values, based on the assumption (called Individual-Global-Max) of convergence to the optimal joint deterministic policy. Markov Shapley value proposed in this paper belongs to the family of value factorisation, based on the game-theoretical framework called MCG that enjoys the interpretability. From the conventional cooperative games (e.g., network flow game [29], induced subgraph game [30] that can be used for modelling social networks, and facility location game [31]), it is insightful that the coalition introduced in this paper exists. In many scenarios, however, the information of coalition might be unknown. Therefore, the latent coalition is assumed, and we only need to concentrate on the observable information, e.g., the global reward.

**Relationship to VDN.** By setting $\delta_i(\mathbf{s}, a_i) = 1$ for all state-action pairs, SHAQ degrades to VDN [8]. Although VDN tried to tackle the problem of dummy agents, Sunehag et al. [8] did not give a theoretical guarantee on identifying it. The Markov Shapley value theory proposed in this paper well addresses this issue from both theoretical and empirical aspects. These aspects show that VDN is a subclass of SHAQ. The theoretical framework proposed in this paper answers to why VDN works well in most scenarios but performs poorly in some scenarios (i.e., $\delta_i(\mathbf{s}, a_i) = 1$ in Eq.9 was incorrectly defined over the suboptimal actions).

**Relationship to COMA.** Compared with COMA [7], each agent $i$'s credit assignment $\bar{Q}_i(\mathbf{s}, a_i)$ is mathematically expressed as follows:

$$\bar{Q}_i(\mathbf{s}, a_i) = \bar{Q}^\pi(\mathbf{s}, \mathbf{a}) - \bar{Q}^{\pi_{-i}}(\mathbf{s}, \mathbf{a}_{-i}),$$
$$\bar{Q}^{\pi_{-i}}(\mathbf{s}, \mathbf{a}_{-i}) = \sum_{a_i} \pi_i(a_i | \mathbf{s}) \bar{Q}^\pi(\mathbf{s}, (\mathbf{a}_{-i}, a_i)),$$

where subscript $-i$ indicates the agents excluding $i$. $\bar{Q}_i(\mathbf{s}, a_i)$ can be seen as the action marginal contribution between the grand coalition Q-value and the coalition Q-value excluding the agent $i$,

under *some permutation to form the grand coalition* wherein agent $i$ is located at the *last position*. The efficiency is obviously violated (i.e., the sum of optimal action marginal contributions defined here is unlikely to be equal to the optimal grand coalition Q-value). In contrast to COMA, SHAQ considers all permutations to form the grand coalition to preserve the efficiency.

**Relationship to Independent Learning.** Independent learning (e.g. IQL [26]) can be also seen as a special credit assignment, however, the credit assigned to each agent is still with no intuitive interpretation. Mathematically, suppose that $\bar{Q}_i(\mathbf{s}, a_i)$ is the independent Q-value of agent $i$, we can rewrite it in the form consisting of action marginal contributions such that

$$\bar{Q}_i(\mathbf{s}, a_i) = \mathbb{E}_{\mathcal{C}_i \sim Pr(\mathcal{C}_i|\mathcal{N}\setminus\{i\})}\left[\bar{\Phi}_i(\mathbf{s}, a_i|\mathcal{C}_i)\right].$$

It is intuitive to see that the independent Q-value is a direct approximation of MSQ, ignoring coalition formation, while SHAQ considers coalition formation in approximation. This gives an explanation for why independent learning works well in some cooperative tasks [32]. Nevertheless, it encounters the same issue as in COMA, the loss of properties led by the coalition formation.

**Relationship to SQDDPG.** We now discuss the relationship between SQDDPG [13] and SHAQ. In terms of algorithms, SQDDPG belongs to policy gradient methods (i.e. an approximation of policy iteration) while SHAQ belongs to value based methods (i.e. an approximation of value iteration). Since policy iteration (with one-step policy evaluation) is equivalent to value iteration [33] (at least under a finite state space and a finite action space), the theory behind SHAQ directly *fills the gap in SQDDPG on theoretical guarantees of convergence to optimal joint policy*. Specifically, the learning procedure of SQDDPG iteratively performs the following two stages:

**Stage 1:** $\quad \min_{\theta} \mathbb{E}_{\mathbf{s},\mathbf{a},R,\mathbf{s}'}\left[\left(R + \gamma \sum_{i\in\mathcal{N}} \hat{Q}_i^{\phi}(\mathbf{s}', a_i'; \theta^-) - \sum_{i\in\mathcal{N}} \hat{Q}_i^{\phi}(\mathbf{s}, a_i; \theta)\right)^2\right].$

**Stage 2:** $\quad \pi_i(\mathbf{s}) \in \arg\max_{a_i} \hat{Q}_i^{\phi}(\mathbf{s}, a_i; \theta).$

It can be observed that both SQDDPG and SHAQ ideally converge to the same optimal MSQs w.r.t. the optimal actions such that

$$\mathbb{E}_{\mathbf{s},\mathbf{s}'}\left[\left(\max_{\mathbf{a}} R(\mathbf{s}, \mathbf{a}) + \gamma \sum_{i\in\mathcal{N}} \max_{a_i'} \hat{Q}_i^{\phi^*}(\mathbf{s}', a_i') - \sum_{i\in\mathcal{N}} \max_{a_i} \hat{Q}_i^{\phi^*}(\mathbf{s}, a_i)\right)^2\right] = 0.$$

However, about suboptimal actions, *SQDDPG does not provide any theoretical guarantee*, whereas SHAQ does with specific implementations as shown in Eq.13 to match the theoretical results shown in this paper. Note that this is critical to reliable interpretations of the optimal MSQ w.r.t. suboptimal actions (e.g., for detecting adversarial attacks on controllers if deployed in industry [34]).

# 6 Experiments

In this section, we show the experimental results of SHAQ on Predator-Prey [17] and various tasks in StarCraft Multi-Agent Challenge (SMAC) [2]. The baselines that we select for comparison are COMA [7], VDN [8], QMIX [9], MASAC [36], QTRAN [10], QPLEX [37] and W-QMIX (including CW-QMIX and OW-QMIX) [35]. The implementation details of our algorithm are shown in Appendix B.1, whereas the implementation of baselines are from [35] [3]. We also compare SHAQ with SQDDPG [13] [4], which is shown in Appendix C.3. For all experiments, we use the $\epsilon$-greedy exploration strategy, where $\epsilon$ is annealed from 1 to 0.05. The annealing time steps vary among different experiments. For Predator-Prey, we apply 1 million time steps for annealing, following the setup from [37]. For the easy and hard maps in SMAC, we apply 50k time steps for annealing, the same as that in [18]; while for the super-hard maps in SMAC, we apply 1 million time steps for annealing to obtain more explorations so that more state-action pairs can be visited. About the replay buffer size, we set as 5000 for all algorithms that is the same as that in [35]. To fairly evaluate all algorithms, we run each experiment with 5 random seeds. All graphs showing experimental results are plotted with the

---

[2] The version that we use in this paper is SC2.4.6.2.69232 rather than the newer SC2.4.10. As reported from [35], the performance is not comparable across versions.

[3] The source code of baseline implementation is from `https://github.com/oxwhirl/wqmix`.

[4] The code of SQDDPG is implemented based on `https://github.com/hsvgbkhgbv/SQDDPG`.

median and 25%-75% quartile shading. About the interpretability of algorithms, we evaluate the algorithms with both both $\epsilon$-greedy policy (i.e., $\epsilon = 0.8$) for obtaining mixed optimal and suboptimal actions and greedy policy for obtaining pure optimal actions. The ablation study of SHAQ is shown in Appendix C.4.

## 6.1 Predator-Prey

We firstly run the experiments on a partially-observable task called Predator-Prey [17], wherein 8 predators that are feasible to be controlled aim to capture 8 preys with random policies in a 10x10 grid world. Each agent's observation is a 5x5 sub-grid centering around it. If a prey is captured by coordination of 2 agents, predators will be rewarded by 10. On the other hand, each unsuccessful attempt by only 1 agent will be punished by a negative reward p. In this experiment, we study the behaviors of each algorithm under different values of p (that describes different levels of coordination). As [35] reported, only QTRAN and W-QMIX can solve this task, while [37] found that the failure was primarily due to the lack of explorations. As a result, we apply the identical epsilon annealing schedule (i.e. 1 million time steps) adopted in [37].

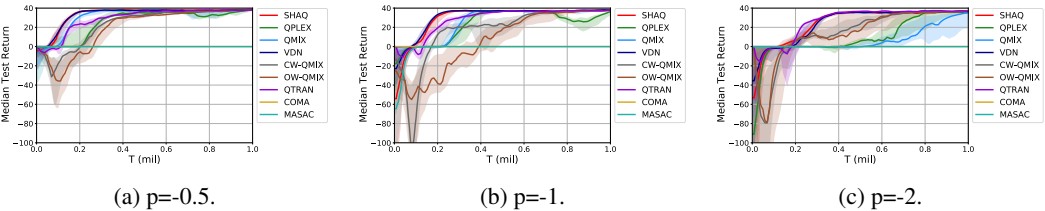

(a) p=-0.5.            (b) p=-1.            (c) p=-2.

Figure 1: Median test return for Predator-Prey with different values of p.

**Performance Analysis.** As Figure 1 shows, SHAQ can always solve the tasks with different values of p. With the epsilon annealing strategy from [37], W-QMIX does not perform as well as reported in [35]. The reason could be its poor robustness to the increased explorations [35] for this environment (see the evidential experimental results in Appendix C.6). The good performance of VDN validates our analysis in Section 5, whereas the performance of QTRAN is surprisingly almost invariant to the value of p. The performances of QPLEX and QMIX become obviously worse when p=-2. The failure of MASAC and COMA could be due to that relative overgeneralisation[5] prevents policy gradient methods from better coordination [39].

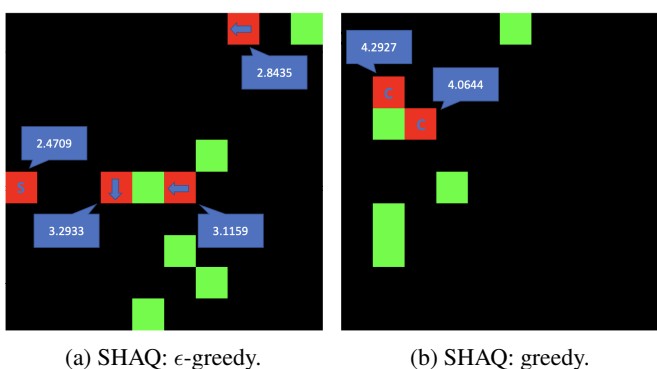

(a) SHAQ: $\epsilon$-greedy.            (b) SHAQ: greedy.

Figure 2: Visualisation of the evaluation for SHAQ on Predator-Prey: each red square is a controllable agent, whereas each green square indicates a prey. Each agent's factorised Q-value is reported in the bubble in blue and the symbols within the squares indicate the action of each agent (i.e., arrows imply the movement direction, "S" implies staying and "C" implies capturing a prey that is valid only when the agent is around a prey).

**Interpretability of SHAQ.** To verify that SHAQ possesses the interpretability, we show its credit assignment on Predator-Prey. As we see from Figure 2b, all agents are around and capture a prey, so

---

[5]Relative overgeneralisation is a common game theoretic pathology that the suboptimal actions are preferred when matched with arbitrary actions from the collaborating agents [38].

both of them perform the optimal actions and deserve almost the equal optimal credit assignment as 4.2927 and 4.0644, which verifies our theoretical claim. From Figure 2a, it can be seen that two agents are far away from preys, so they receive low credits as 2.4709 and 2.8435. On the other hand, the other two agents are around a prey, but they do not perform the optimal action "capture", so they receive less credits than the two agents in Figure 2b. Nevertheless, they are around a prey, so they perform better than those agents that are far away from preys and receive comparatively greater credits as 3.2933 and 3.1159. The coherent credit assignments in both Figure 2a and 2b implies that the assigned credits reflect agents' contributions (verifying (iii) in Proposition 2) , i.e., each agent receives the credit that is consistent with its decision.

## 6.2 StarCraft Multi-Agent Challenge

We next evaluate SHAQ on the more challenging SMAC tasks, the environmental settings of which are the same as that in [18]. To broadly compare the performance of SHAQ with baselines, we select 4 easy maps: 8m, 3s5z, 1c3s5z and 10m_vs_11m; 3 hard maps: 5m_vs_6m, 3s_vs_5z and 2c_vs_64zg; and 4 super-hard maps: 3s5z_vs_3s6z, Corridor, MMM2 and 6h_vs_8z. All training is through online data collection. Due to the limited space, we only show partial results in the main part of paper and leave the rest in Appendix C.1.

**Performance Analysis.** It shows in Figure 3 that SHAQ outperforms all baselines on all maps, except for 6h_vs_8z. On 6h_vs_8z, SHAQ can beat all baselines except for CW-QMIX. VDN performs well on 4 maps but bad on the other 2 maps, which still verifies our analysis in Section 5. QMIX and QPLEX perform well on the most of maps, except for 3s_vs_5z, 2c_vs_64zg and 6h_vs_8z. As for COMA, MADDPG and MASAC, their poor performances could be due to the weak adaptability to challenging tasks. Although QTRAN can theoretically represent the complete class of the global Q-value [10], its complicated learning paradigm could impede the convergence to the value function for challenging tasks and therefore result in the poor performance. Although W-QMIX performs well on some maps, owing to lacking a law on hyperparameter tuning [35] it is difficult to be adapted for all scenarios (see Appendix C.2).

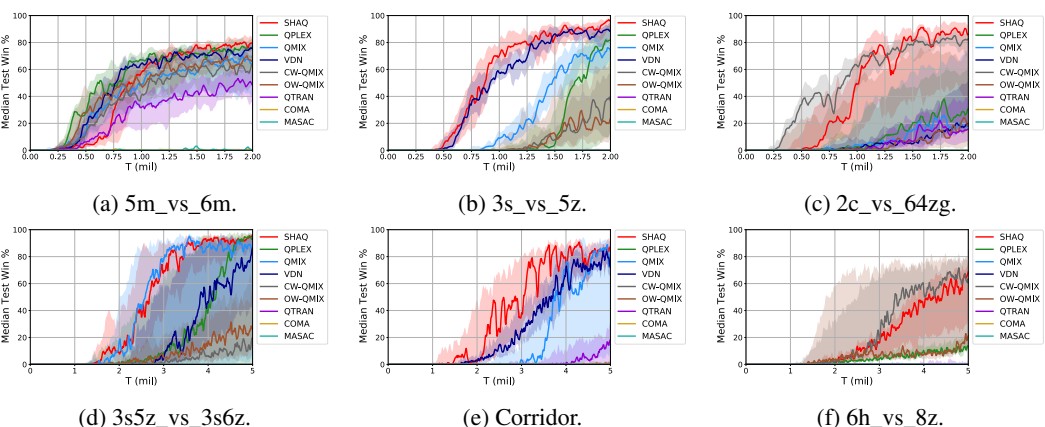

Figure 3: Median test win % for hard (a-c), and super-hard (d-f) maps of SMAC.

**Interpretability of SHAQ.** To further show the interpretability of SHAQ, we also conduct a test on 3m (i.e. a simple task in SMAC). As seen from Figure 4a, Agent 3 faces the direction opposite to enemies, meanwhile, the enemies are out of its attacking range. It can be understood as that Agent 3 does not contribute to the team and thus it is almost a dummy agent. Its MSQ is 0.84 (around 0) that correctly catch the manner of a dummy agent (verifying (i) in Proposition 2). In contrast, Agent 1 and Agent 2 are attacking enemies, while Agent 1 suffers from more attacks (with lower health) than Agent 2. As a result, Agent 1 contributes more than Agent 2 and therefore its MSQ is greater, which implies that the credits reflect agents' contributions (verifying (iii) in Proposition 2). On the other hand, we can see from Figure 4e that with the optimal policies all agents receive almost identical MSQs (verifying the theoretical results in Section 4.1). The above results well verify the theoretical analysis that we deliver before.

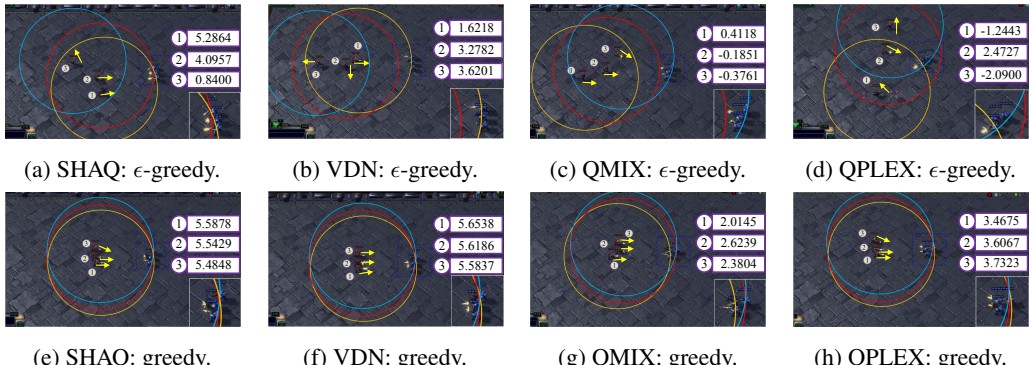

| (a) SHAQ: $\epsilon$-greedy. | (b) VDN: $\epsilon$-greedy. | (c) QMIX: $\epsilon$-greedy. | (d) QPLEX: $\epsilon$-greedy. |
|---|---|---|---|
| (e) SHAQ: greedy. | (f) VDN: greedy. | (g) QMIX: greedy. | (h) QPLEX: greedy. |

Figure 4: Visualisation of the test for SHAQ and baselines on 3m in SMAC: each colored circle is the centered attacking range of a controllable agent (in red), and each agent's factorised Q-value is reported on the right. We mark the direction that each agent face by an arrow for clearness.

To justify that the MSQs learned by SHAQ are non-trivial, we also show the results of VDN, QMIX and QPLEX. It is surprising that the Q-values of these baselines are also almost identical among agents for the optimal actions (however, the property disappears in more complicated scenarios as shown in Appendix C.5 while the property of SHAQ is still valid). Since VDN is a subclass of SHAQ and possesses the same form of loss function for optimal actions, it is reasonable that it obtains the similar results to SHAQ. As for the suboptimal actions, VDN does not possess an explicit interpretation as SHAQ due to the incorrect definition of $\delta_i(\mathbf{s}, a_i) = 1$ over suboptimal actions (verifying the statement in Section 5). The values of QMIX and QPLEX are difficult to be explained.

## 7   Conclusion

**Summary.** This paper generalises Shapley value to Markov convex game, called Markov Shapley value. Markov Shapley value inherits a number of properties: (i) identifiability of dummy agents; (ii) efficiency; (iii) reflecting the contribution and (iv) symmetry. Based on Property (ii), we derive Shapley-Bellman optimality equation, Shapley-Bellman operator and SHAQ. We prove that solving Shapley-Bellman optimality equation is equivalent to solving the Markov core (i.e., no agent has incentives to deviate from the grand coalition). Markov convex game with the grand coalition is equivalent to global reward game [13], wherein Markov Shapley value plays the role of value factorisation. Since SHAQ is a stochastic approximation of Shapley-Bellman operator that is proved to solve Shapley-Bellman optimality equation, global reward game with value factorisation becomes valid standing by the cooperative game theoretical framework (i.e. solving Markov core). Property (i) and (iii) in Proposition 2 are demonstrated in the experiments showing the interpretability of SHAQ.

**Limitation and Future Work.** The value of Markov convex game is not limited to solving problems with the grand coalition, though in this paper we design SHAQ that only focuses on the scenario with the grand coalition. By removing the condition of supermodularity (see Eq.1), this framework can be used to study more general coalition games where different coalitions of agents as units may compete/cooperate with each other. Since the grand coalition and Markov Shapley value is not a solution in Markov core yet, the learning process becomes more complicated to converge to Markov core. A possible research direction in future is to investigate dynamically forming the coalition structure and conducting credit assignments simultaneously.

## Acknowledgements

This work is sponsored by the Engineering and Physical Sciences Research Council of UK (EPSRC) under awards EP/S000909/1. Tae-Kyun Kim is partly sponsored by KAIA grant (22CTAP-C163793-02, MOLIT), NST grant (CRC 21011, MSIT), KOCCA grant (R2022020028, MCST) and the Samsung Display corporation. Yuan Zhang is sponsored by the European Union's Horizon 2020 research and innovation program under the Marie Skłodowska-Curie grant agreement No. 953348 (ELO-X).

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
