# A  Algorithm of Shapley Q-learning

In this section, we present the pseudo code of Shapley Q-learning in Algorithm 1. The general paradigm can be divided into such parts: (1) collecting samples through $\epsilon$-greedy strategy and store the collected samples to a replay buffer for training; (2) sampling a batch of episodes of samples from the replay buffer; (3) calculating $\hat{Q}_i(\tau_i^{t+1}, a_i^{t+1}; \theta^-)$, $\hat{\alpha}_i(\mathbf{s}, a_i^k; \lambda)$ and $\hat{Q}_i(\tau_i^t, a_i^t; \theta)$; and (4) constructing a loss of Shapley Q-learning and updating parameters to minimise the loss.

---

**Algorithm 1** Shapley Q-learning

---

1: Initialise a set of agents $\mathcal{N}$ and set $N = |\mathcal{N}|$
2: Initialise $\hat{Q}_i(\tau_i, a_i; \theta)$ with the shared parameters among agents
3: Initialise $\hat{\alpha}_i(\mathbf{s}, a_i; \lambda)$ with the shared parameters among agents
4: Initialise $\hat{Q}_i(\tau_i, a_i; \theta^-)$ by copying $\hat{Q}_i(\tau_i, a_i; \theta)$ with the shared parameters among agents
5: Initialise a replay buffer $\mathcal{B}$
6: **repeat**
7:     Initialise a container $\mathcal{E}$ for storing an episode
8:     Observe an initial global state $\mathbf{s}^1$ and each agent's partial observation $o_i^1$ from an environment
9:     **for** t=1:T **do**
10:         Get $\tau_i^t = (o_i^m)_{m=1:t}$ for each agent
11:         For each agent $i$, select an action

$$a_i^t = \begin{cases} \text{a random action} & \text{with probability } \epsilon \\ \arg\max_{a_i} \hat{Q}_i^*(\tau_i^t, a_i; \theta) & \text{otherwise} \end{cases}$$

12:         Execute $a_i^t$ of each agent to get the global reward $R^t$, $\mathbf{s}^{t+1}$ and each agent's $o_i^{t+1}$
13:         Store $\langle \mathbf{s}^t, (o_i^t)_{i=1:N}, (a_i^t)_{i=1:N}, R^t, \mathbf{s}^{t+1}, (o_i^{t+1})_{i=1:N} \rangle$ to $\mathcal{E}$
14:     **end for**
15:     Store $\mathcal{E}$ to $\mathcal{B}$
16:     Sample a batch of episodes with batch size B from $\mathcal{B}$
17:     **for** each sampled episode **do**
18:         **for** k=1:T **do**
19:             Get each transition $\langle \mathbf{s}^k, (o_i^k)_{i=1:N}, (a_i^k)_{i=1:N}, R^k, \mathbf{s}^{k+1}, (o_i^{k+1})_{i=1:N} \rangle$
20:             For each agent $i$, get $\tau_i^k = (o_i^m)_{m=1:k}$
21:             For each agent $i$, calculate $\hat{Q}_i(\tau_i^k, a_i^k; \theta)$
22:             For each agent $i$, calculate $\alpha_i(\mathbf{s}^k, a_i^k; \lambda)$ by Algorithm 2
23:             For each agent $i$, calculate $\delta_i(\mathbf{s}^k, a_i^k; \lambda)$ as follows:

$$\hat{\delta}_i(\mathbf{s}^k, a_i^k; \lambda) = \begin{cases} 1 & a_i^k = \arg\max_{a_i} \hat{Q}_i(\mathbf{s}^k, a_i; \theta) \\ \hat{\alpha}_i(\mathbf{s}^k, a_i^k; \lambda) & a_i^k \neq \arg\max_{a_i} \hat{Q}_i(\mathbf{s}^k, a_i; \theta) \end{cases} \text{ (via Algorithm 2)}$$

24:             For each agent $i$, get $\tau_i^{k+1} = (o_i^m)_{m=1:k+1}$
25:             For each agent $i$, get $a_i^{k+1}$ by $\arg\max_{a_i} \hat{Q}_i(\tau_i^{k+1}, a_i; \theta)$
26:             For each agent $i$, calculate $\hat{Q}_i(\tau_i^{k+1}, a_i^{k+1}; \theta^-)$
27:         **end for**
28:     **end for**
29:     Construct a loss as follows:

$$\min_{\theta, \lambda} \frac{1}{B} \sum_{k=1}^{B} \left[ \left( R^k + \gamma \sum_{i \in \mathcal{N}} \max_{a_i^k} \hat{Q}_i(\tau_i^{k+1}, a_i^{k+1}; \theta^-) - \sum_{i \in \mathcal{N}} \hat{\delta}_i(\mathbf{s}^k, a_i^k; \lambda) \, \hat{Q}_i(\tau_i^k, a_i^k; \theta) \right)^2 \right]$$

30:     Update $\theta$ and $\lambda$ through the above loss
31:     Periodically update $\theta^-$ by copying $\theta$
32: **until** $\hat{Q}_i(\tau_i, a_i; \theta)$ converges

---

**Implementation of Sampling from $Pr(\mathcal{C}_i | \mathcal{N} \backslash \{i\})$ (Line 4 in Algorithm 2).** As introduced in Remark 1, the analytic form of $Pr(\mathcal{C}_i | \mathcal{N} \backslash \{i\})$ is $\frac{|\mathcal{C}_i|!(|\mathcal{N}| - |\mathcal{C}_i| - 1)!}{|\mathcal{N}|!}$ that is actually the occurrence frequency of correlated coalition $\mathcal{C}_i$. Since each coalition is formed by different permutations, it can be instead sampled from permutations directly with uniform distribution where $\frac{1}{|\mathcal{N}|!}$ is as the probability distribution over each permutation. It is not difficult to find that these two sampling strategy induce the same probability distribution for obtaining $\mathcal{C}_i$, so they are equivalent. In practice, we sample multiple permutations (saying $M$) from the uniform distribution in parallel. From each sampled permutation, we extract the the relevant $\mathcal{C}_i$ for each agent $i$. Afterwards, to each agent $i$, $M$ coalitions are obtained to calculate $\hat{\alpha}_i(\mathbf{s}, a_i)$.

**Algorithm 2** Calculating $\hat{\alpha}_i(\mathbf{s}, a_i)$

---

1: **Input:** $\mathbf{s}, \left(\hat{Q}_i(\tau_i, a_i; \theta)\right)_{i=1:N}, M$
2: **Output:** $\left(\hat{\alpha}_i(\mathbf{s}, a_i)\right)_{i=1:N}$
3: **for** each agent $i$ **do**
4:     Sample $M$ preceding coalitions $\mathcal{C}_i^k \sim Pr(\mathcal{C}_i | \mathcal{N} \backslash \{i\})$
5:     **for** k=1:M **do**
6:         Get $\hat{Q}_{\mathcal{C}_i^k}(\tau_{\mathcal{C}_i^k}, \mathbf{a}_{\mathcal{C}_i^k}) = \frac{1}{|\mathcal{C}_i^k|} \sum_{j \in \mathcal{C}_i^k} \hat{Q}_j(\tau_j, a_j)$
7:     **end for**
8:     Get $\hat{\alpha}_i(\mathbf{s}, a_i) = \frac{1}{M} \sum_{k=1}^{M} F_{\mathbf{s}}\left(\hat{Q}_{\mathcal{C}_i^k}(\tau_{\mathcal{C}_i^k}, \mathbf{a}_{\mathcal{C}_i^k}), \hat{Q}_i(\tau_i, a_i)\right) + 1$
9: **end for**

---

# B  Experimental Setups

## B.1  Implementation Details of Shapley Q-learning

We now provide the additional implementation details that are omitted from the main part of paper. First, $F_s(\cdot, \cdot)$ is a 3-layer network (consecutively with two affine transformation and an activation of absolute), where the hidden-layer dimension is 32. The parameters of each affine transformation are generated by hyper-networks [40] with input as the global state, whose details are shown in Table 1. The architecture of each agent's Q-value is a RNN with GRUs cell [23], whose hidden-layer dimension is 64. The input dimension is state dimension and the output dimension is action dimension.

Table 1: Table of specifications for $F_s(\cdot, \cdot)$.

| NETWORK | STRUCTURE |
|---|---|
| 1ST WEIGHT MATRIX | [ LINEAR(STATE_DIM, 64), RELU, LINEAR(64, 32*2), ABSOLUTE ] |
| 1ST BIAS | [ LINEAR(STATE_DIM, 64) ] |
| 2ND WEIGHT MATRIX | [ LINEAR(STATE_DIM, 64), RELU, LINEAR(64, 32), ABSOLUTE ] |
| 2ND BIAS | [ LINEAR(STATE_DIM, 32), RELU, LINEAR(32, 1) ] |

Taking the lessons of training two coupling modules from GANs [41], we take separate learning rates for $\hat{\alpha}_i(\mathbf{s}, a_i)$ and $\hat{Q}_i(\mathbf{s}, a_i)$. The learning rate for $\hat{Q}_i(\mathbf{s}, a_i)$ is fixed at 0.0005 for all tasks. Nevertheless, the learning rate for $\hat{\alpha}_i(\mathbf{s}, a_i)$ is dependent on the number of controllable agents. We use RMSProp optimizer for training in all tasks. All models are implemented in PyTorch 1.4.0 and each experiment is run on Nvidia GeForce RTX 2080Ti for 4 to 26 hours with a single process of environment.

## B.2  Hyperparameters of Baselines

The hyperparameters of all baselines except for SQDDPG [13] are consistent with Rashid et al. [35] and Wang et al. [37]. The hyperparamers of SQDDPG are shown as follows: (1) The policy network is consistent with the other baselines, while the critic network is with 3 hidden layers and each layer is with 64 neurons. (2) The policy network is updated every 2 time steps, while the critic network is updated each time step. (3) The multiplier of the entropy of policy is 0.005. The rest of settings are identical with other baselines.

## B.3  Predator-Prey for Modelling Relative Overgeneralisation

We give the experimental setups of Predator-Prey [17] in Table 2.

## B.4  StarCraft Multi-Agent Challenge

The StarCraft Multi-Agent Challenge (SMAC) [18] is a popular testbed for multi-agent reinforcement learning (MARL) algorithms. The main difficulties are (1) challenging dynamics, (2) partial observability and (3) high-dimensional observation space. During training, both the global state of the environment and each agent's local observation are able to be obtained; however, during execution,

Table 2: Table of experimental setups of Predator-Prey.

| Hyperparameters | Value | Description |
|---|---|---|
| Batch size | 32 | The number of episodes for each update |
| Discount factor $\gamma$ | 0.99 | The importance of future rewards |
| Replay buffer size | 5,000 | The maximum number of episodes to store in memory |
| Episode length | 200 | Maximum time steps per episode |
| Test episode | 16 | The number of episodes for evaluating the performance |
| Test interval | 10,000 | The time step frequency for evaluating the performance |
| Epsilon start | 1.0 | The start epsilon $\epsilon$ value for exploration |
| Epsilon finish | 0.05 | The final epsilon $\epsilon$ value for exploration |
| Exploration step | 1,000,000 | The number of steps for linearly annealing $\epsilon$ |
| Max training step | 1,000,000 | The number of training steps |
| Target update interval | 200 | The update frequency for target network |
| Learning rate | 0.0001 | The learning rate for $\delta_i(\mathbf{s}, a_i)$ |
| $\alpha$ for W-QMIX variants | 0.1 | The weight for CW-QMIX and OW-QMIX |
| Sample size | 10 | The sample size for coalition sampling |

Table 3: Introduction of maps and characters in SMAC.

| Map Name | Ally Units | Enemy Units | Categories |
|---|---|---|---|
| 3s5z | 3 Stalkers & 5 Zealots | 3 Stalkers & 5 Zealots | Easy |
| 1c3s5z | 1 Colossi & 3 Stalkers & 5 Zealots | 1 Colossi & 3 Stalkers & 5 Zealots | Easy |
| 8m | 8 Marines | 8 Marines | Easy |
| 10m_vs_11m | 10 Marines | 11 Marines | Easy |
| 5m_vs_6m | 5 Marines | 6 Marines | Hard |
| 3s_vs_5z | 3 Stalkers | 5 Zealots | Hard |
| 2c_vs_64zg | 2 Colossi | 64 Zerglings | Hard |
| 3s5z_vs_3s6z | 3 Stalkers & 5 Zealots | 3 Stalkers & 6 Zealots | Super-Hard |
| MMM2 | 1 Medivac, 2 Marauders & 7 Marines | 1 Medivac, 3 Marauders & 8 Marines | Super-Hard |
| 6h_vs_8z | 6 Hydralisks | 8 Zerglings | Super-Hard |
| Corridor | 6 Zealots | 24 Zerglings | Super-Hard |

only each agent's local observation can be observed. For this reason, SMAC fits the centralised training and decentralised execution (CTDE) paradigm. In each micromanagement task, the ally units are controlled by agents and the enemy units are controlled by the built-in game AI. The agents need to learn a strategy to solve some challenging combat scenarios and defeat their opponents with maximum win rate.

In this paper, we evaluate the proposed SHAQ on 11 typical combat scenarios in SMAC that can be classified into three categories: easy (8m, 3s5z, 1c3s5z and 10m_vs_11m), hard (5m_vs_6m, 3s_vs_5z and 2c_vs_64zg), and super-hard (3s5z_vs_3s6z, Corridor, MMM2 and 6h_vs_8z). More details of these tasks are provided in Table 3. The specific experimental setups for SMAC are shown in Table 4 and 5.

Table 4: Table of experimental setups for SMAC.

| Hyperparameters | Easy | Hard | Super Hard | Description |
|---|---|---|---|---|
| Batch size | 32 | 32 | 32 | The number of episodes for each update |
| Discount factor $\gamma$ | 0.99 | 0.99 | 0.99 | The importance of future rewards |
| Replay buffer size | 5,000 | 5,000 | 5,000 | The maximum number of episodes to store in memory |
| Max training step | 2,000,000 | 2,000,000 | 5,000,000 | The number of training steps |
| Test episode | 32 | 32 | 32 | The number of episodes for evaluation |
| Test interval | 10,000 | 10,000 | 10,000 | The time step frequency for evaluating the performance |
| Epsilon start | 1.0 | 1.0 | 1.0 | The start epsilon $\epsilon$ value for exploration |
| Epsilon finish | 0.05 | 0.05 | 0.05 | The final epsilon $\epsilon$ value for exploration |
| Exploration step | 50,000 | 50,000 | 1,000,000 | The number of steps for linearly annealing $\epsilon$ |
| Target update interval | 200 | 200 | 200 | The update frequency for target network |
| $\alpha$ for OW-QMIX | 0.5 | 0.5 | 0.5 | The weight for OW-QMIX |
| $\alpha$ for CW-QMIX | 0.75 | 0.75 | 0.75 | The weight for CW-QMIX |
| Sample size | 10 | 10 | 10 | The sample size for coalition sampling |

Table 5: The learning rate for training $\hat{\alpha}_i(\mathbf{s}, a_i)$ of SHAQ for various maps in SMAC.

| MAP NAME | NUMBER OF AGENTS | LEARNING RATE FOR $\hat{\alpha}_i(\mathbf{s}, a_i)$ |
|---|---|---|
| 2C_VS_64ZG | 2 | 0.002 |
| 3S_VS_5Z | 3 | 0.001 |
| 5M_VS_6M | 5 | 0.0005 |
| 6H_VS_8Z | 6 | 0.0005 |
| CORRIDOR | 6 | 0.0005 |
| 8M | 8 | 0.0003 |
| 3S5Z | 8 | 0.0003 |
| 3S5Z_VS_3S6Z | 8 | 0.0003 |
| 1C3S5Z | 9 | 0.0002 |
| 10M_VS_11M | 10 | 0.0001 |
| MMM2 | 10 | 0.0001 |

## C    Extra Experimental Results

### C.1    Experimental Results on Extra SMAC Maps

To thoroughly compare the performance of SHAQ with baselines, we also run experiments on 5 extra maps in SMAC as Figure 5 shows. 8m, 3s5z, 1c3s5z and 10m_vs_11m are an easy maps and MMM2 is a super-hard map. The strategy of epsilon annealing is consistent with the previous experiments for SMAC. It is obvious that SHAQ also performs generally well on these 5 maps.

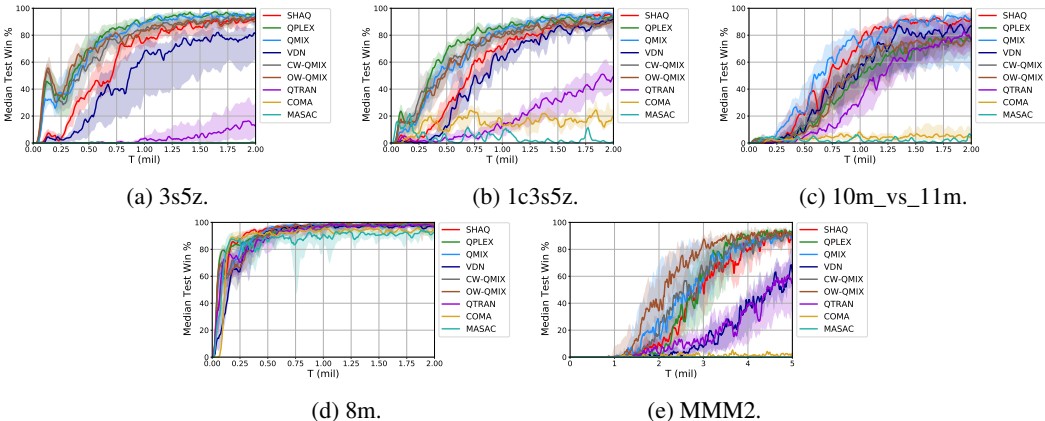

(a) 3s5z.    (b) 1c3s5z.    (c) 10m_vs_11m.

(d) 8m.    (e) MMM2.

Figure 5: Median test win % for 5 extra maps in SMAC.

### C.2    Extra Experimental Results on W-QMIX with $\alpha = 0.1$

To show the significance of tuning $\alpha$ for W-QMIX, we also run W-QMIX with $\alpha = 0.1$ in addition to the best $\alpha$ reported in [35]. We can observe from Figure 6 that the performances of W-QMIX are not comparatively identical for each choice of $\alpha$. As a result, W-QMIX suffers from the separate tuning of $\alpha$ for each scenario. Unfortunately, Rashid et al. [35] did not provide an empirical law for selecting $\alpha$, while SHAQ enjoys an empirical law to select $\hat{\alpha}_i(\mathbf{s}, a_i)$ as Figure 8b shows.

### C.3    Comparison with SQDDPG

To emphasize the improvement of SHAQ from SQDDPG [13], we exclusively compare these two algorithms on 3 maps in SMAC. As Figure 7 shows, the performance of SHAQ surpasses that of SQDDPG on all 3 maps, while SQDDPG can only learn on the simplest map 3m. The most possible reason for the failure of SQDDPG to complicated tasks is its sample complexity inefficiency for permutations of agents as discussed in Section 5 that leads to the difficulty in learning. Apparently, the implementation of coalition invariance of SHAQ mitigates this weakness so that it is able to solve

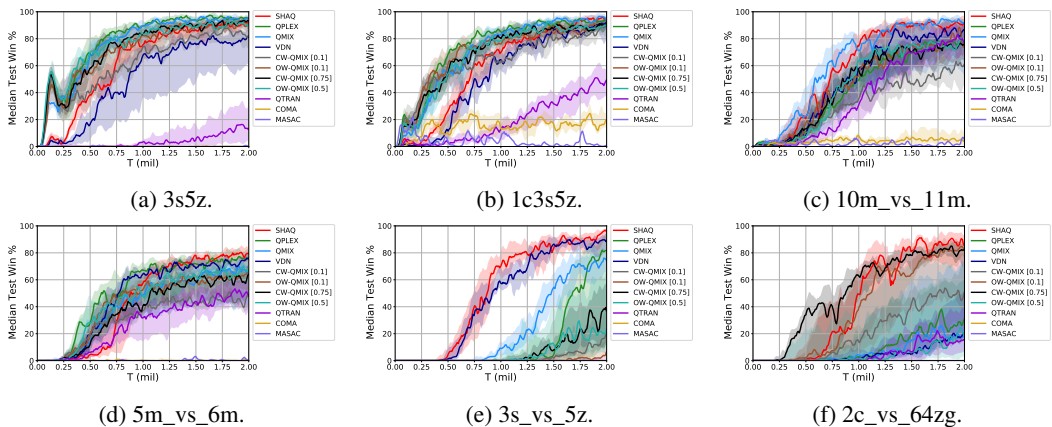

(a) 3s5z.     (b) 1c3s5z.     (c) 10m_vs_11m.

(d) 5m_vs_6m.     (e) 3s_vs_5z.     (f) 2c_vs_64zg.

Figure 6: Median test win % for easy (1st row) and hard (2nd row) maps of SMAC for W-QMIX with different $\alpha$.

more challenging tasks. We also show the results for SQDDPG on Predator-Prey with the same setups (i.e., the epsilon annealing steps are 1 mil), as Figure 10a shows. It is apparent that SHAQ can still outperform SQDDPG.

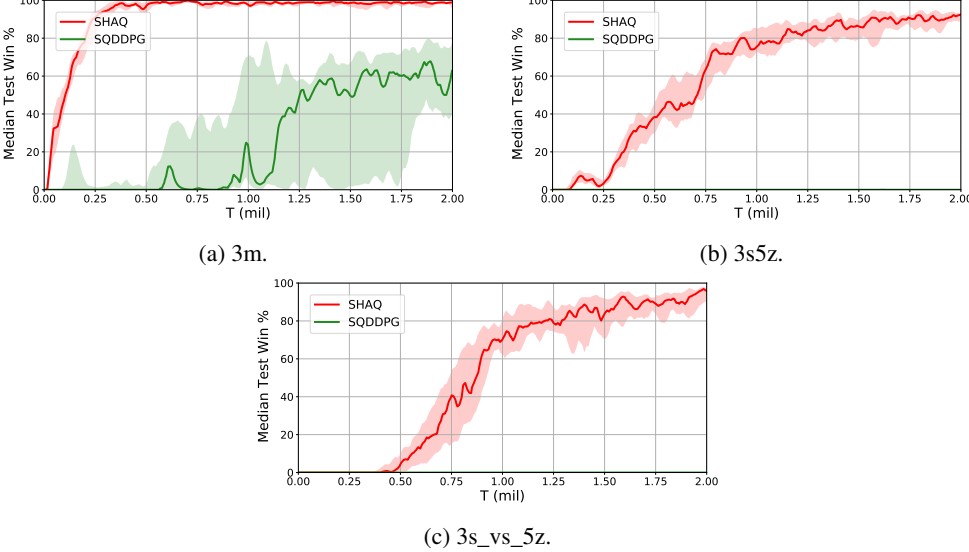

(a) 3m.     (b) 3s5z.

(c) 3s_vs_5z.

Figure 7: Median test win % for 3 maps of SMAC to compare SHAQ with SQDDPG.

### C.4 Ablation Study

We also conduct ablation study of SHAQ, such as the sample size M for approximating $\hat{\alpha}_i(\mathbf{s}, a_i)$, the empirical selection law on the learning rate of $\hat{\alpha}_i(\mathbf{s}, a_i)$, and the demonstration of the necessity of learning $\hat{\alpha}_i(\mathbf{s}, a_i)$ rather than manual setting. These results show that SHAQ is an easy-to-use algorithm that is potential to be applied to other scenarios with less efforts on tuning hyperparameters.

**Sample Size M for Approximating** $\hat{\alpha}(\mathbf{s}, a_i)$**.** To study the impact of sample size M on the performance of SHAQ, we conduct an ablation study as Figure 8a shows. We observe that the small M is able to achieve fast convergence rate but with high variance, while the large M is with low variance but comparatively slow convergence rate. The observations are consistent with the conclusions from stochastic optimisation [42, 43]. As a result, we select M = 10 in practice, to trade off between convergence rate and variance.

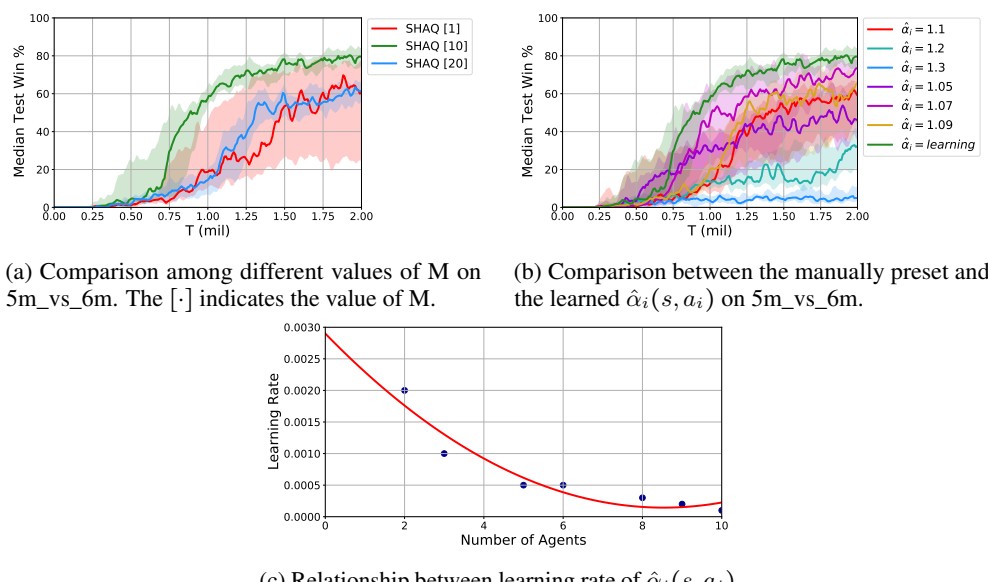

(a) Comparison among different values of M on 5m_vs_6m. The [·] indicates the value of M.

(b) Comparison between the manually preset and the learned $\hat{\alpha}_i(s, a_i)$ on 5m_vs_6m.

(c) Relationship between learning rate of $\hat{\alpha}_i(s, a_i)$ and the number of agents (in red curve).

Figure 8: The figures of 3 ablation studies of SHAQ on SMAC.

**An Empirical Law on Selecting the Learning Rate of** $\hat{\alpha}_i(s, a_i)$**.** To provide an empirical law on selecting the learning rate of $\hat{\alpha}_i(s, a_i)$, we statistically fit a curve of the learning rate w.r.t. the number of controllable agents by the experimental results on SMAC that is shown in Figure 8c. It is seen that the learning rate of $\hat{\alpha}_i(s, a_i)$ is generally negatively related to the number of agents. In other words, as the number of agents grows the learning rate of $\hat{\alpha}_i(s, a_i)$ is recommended to be smaller. For example, if the number of agents is more than 10, the learning rate of $\hat{\alpha}_i(s, a_i)$ is recommended to be 0.0001 as the guidance from Figure 8c.

**The Necessity of Learning** $\hat{\alpha}_i(\mathbf{s}, a_i)$**.** Some readers may be concerned about the necessity of learning $\hat{\alpha}_i(\mathbf{s}, a_i)$. To answer this question, we study the necessity of learning $\hat{\alpha}_i(\mathbf{s}, a_i)$ on 5m_vs_6m. Since the learned $\hat{\alpha}_i(\mathbf{s}, a_i)$ finally converges to 1.1029, we grid search the fixed values of $\hat{\alpha}_i(\mathbf{s}, a_i)$ around this number. As Figure 8b shows, $\hat{\alpha}_i(\mathbf{s}, a_i)$ with manually preset fixed value cannot work as well as the learned $\hat{\alpha}_i(\mathbf{s}, a_i)$. Therefore, we demonstrate the necessity of learning $\hat{\alpha}_i(\mathbf{s}, a_i)$ here.

## C.5 More Visualisation for Interpretability of SHAQ

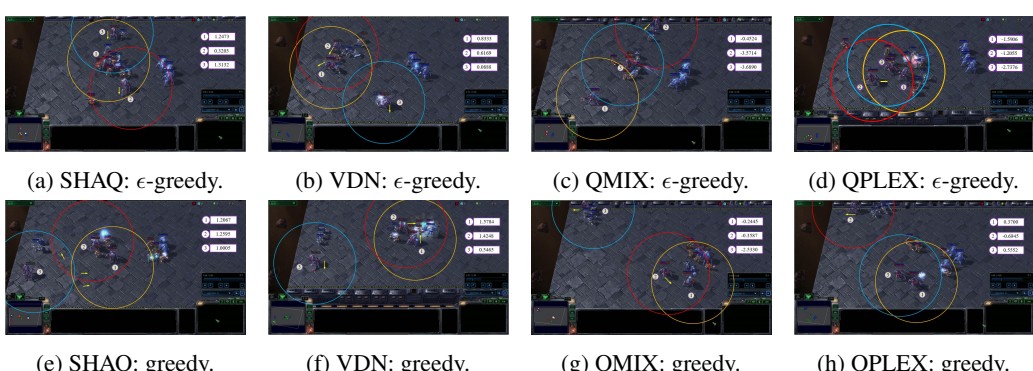

(a) SHAQ: $\epsilon$-greedy.  (b) VDN: $\epsilon$-greedy.  (c) QMIX: $\epsilon$-greedy.  (d) QPLEX: $\epsilon$-greedy.

(e) SHAQ: greedy.  (f) VDN: greedy.  (g) QMIX: greedy.  (h) QPLEX: greedy.

Figure 9: Visualisation of the evaluation for SHAQ and baselines on 3s5z_vs_3s6z in SMAC: each colored circle is the centered attacking range of a controllable agent (in red), and each agent's factorised Q-value is reported on the right. We mark the direction that each moving agent face by an arrow.

To verify our theoretical results more firmly, we show the Q-values on a more complicated scenario in SMAC, i.e. 3s5z_vs_3s6z during test in Figure 9. First, we take a look into the optimal actions. SHAQ can still demonstrate the equal credit assignment as we claimed before. Unfortunately, VDN does not explicitly show equal credit assignment. The possible reason is that part of parameters of Q-value are shared between optimal actions and suboptimal actions. Therefore, the parametric effects of the mistakes conducted on suboptimal actions to the optimal actions by VDN during learning may be exaggerated when the number of agents increases. About QMIX and QPLEX, the Q-values of optimal actions are difficult to be interpreted in this complicated scenario. For both algorithms, the agent who is responsible for kiting [6] (i.e. Agent 3 for QMIX and Agent 2 for QPLEX) receives the lowest credit, however, it is an important role to the team in a combat tactic. Next, we focus on the demonstration of the suboptimal actions. As for SHAQ, Agent 1 and Agent 3 are participating into the battle, so deserving almost the equal credit assignment. However, Agent 2 drops teammates and escapes from the center of battle, so it contributes almost nothing to the team. As a result, it can be seen as a dummy agent and thus obtains the credit near 0. This again agrees with our theoretical analysis. About VDN, it coincidentally receives near 0 for the dummy agent (i.e. Agent 3) in this scenario. Nevertheless, the low credit assignments to the other 2 agents who participate in the battle are difficult to be interpreted. About QMIX, the agents who participate in the battle (i.e. Agent 2 and Agent 3) receive the lowest credits, while the agent (i.e. Agent 1) who escapes from the battle receives the highest credit. For QPLEX, the agents' behaviours are difficult to be interpreted.

### C.6 Extra Experimental Results of Predator-Prey

In Figure 10b and Figure 10c, we show the results of W-QMIX with the annealing steps as 50k to support that the poor performance of W-QMIX on Predator-Prey is due to its poor robustness to the increased explorations.

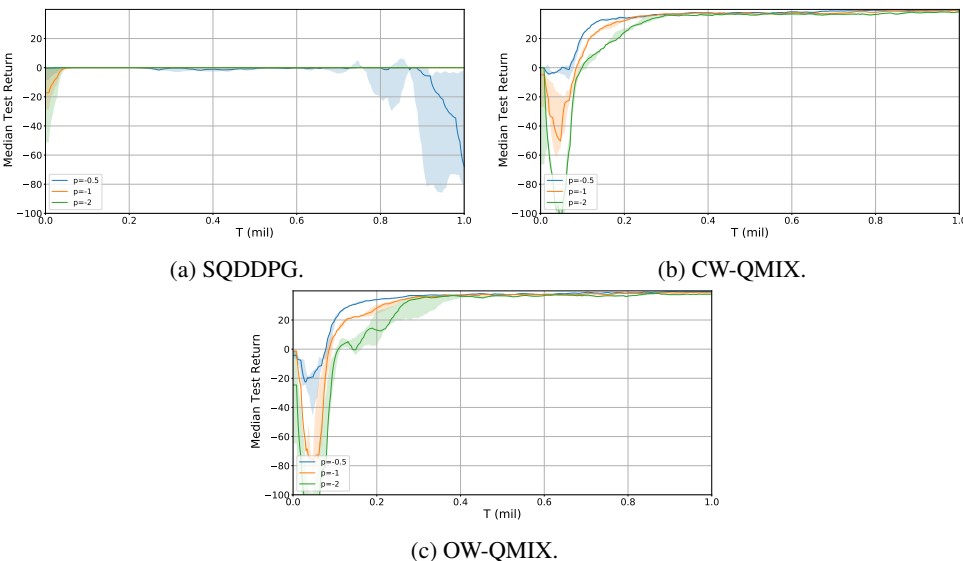

(a) SQDDPG.  (b) CW-QMIX.

(c) OW-QMIX.

Figure 10: Median test return for SQDDPG and W-QMIX (including OW-QMIX and CW-QMIX) on Predator-Prey.

## D  Additional Background

### D.1  Value Factorisation in MARL

Although there are lots of works on value factorisation in MARL, most of them are based on an assumption called Individual-Global-Max (IGM) [10] that is defined in Definition 3.

---

[6] https://en.wikipedia.org/wiki/Glossary_of_video_game_terms.

**Definition 3.** *For a joint Q-value $Q^\pi(\mathbf{s}, \mathbf{a})$ with a deterministic policy, if the following equation is assumed to hold such that*

$$\arg\max_{\mathbf{a}} Q^\pi(\mathbf{s}, \mathbf{a}) = \left( \arg\max_{a_i} Q_i(\mathbf{s}, a_i) \right)_{i=1,2,\ldots,|\mathcal{N}|}, \tag{15}$$

*then we say that $\left( Q_i(\mathbf{s}, a_i) \right)_{i=1,2,\ldots,|\mathcal{N}|}$ satisfies Individual-Global-Max (IGM) and $Q^\pi(\mathbf{s}, \mathbf{a})$ can be factorised by $\left( Q_i(\mathbf{s}, a_i) \right)_{i=1,2,\ldots,|\mathcal{N}|}$.*

There are 3 popular frameworks that are followed by most of works implementing the IGM, called VDN [8], QMIX [9] and QTRAN [10].

**VDN.** VDN linearly factorises a global value function such that

$$Q^\pi(\mathbf{s}, \mathbf{a}) = \sum_{i \in \mathcal{N}} Q_i(\mathbf{s}, a_i), \tag{16}$$

so that Eq.15 holds.

**QMIX.** QMIX learns a monotonic mixing function $f_{\mathbf{s}} : \times_{i \in \mathcal{N}} Q_i(\mathbf{s}, a_i) \times \mathbf{s} \mapsto \mathbb{R}$ to implement the factorisation such that

$$Q^\pi(\mathbf{s}, \mathbf{a}) = f_{\mathbf{s}} \left( Q_1(\mathbf{s}, a_1), \ldots, Q_{|\mathcal{N}|}(\mathbf{s}, a_{|\mathcal{N}|}) \right), \tag{17}$$

so that Eq.15 holds. Although QMIX has a richer functional class of factorisation than that of VDN, it meets a problem that $\max_{\mathbf{a}} Q^\pi(\mathbf{s}, \mathbf{a}) = \sum_{i \in \mathcal{N}} \max_{a_i} Q_i(\mathbf{s}, a_i)$ does not necessarily hold, which may lead to the bias on Q-value estimation [10] and affect the learning process to achieve the optimal joint policy. Theoretically, VDN does not possess the problem discussed above, however, the functional class of the simply additive factorisation is so restrictive [9].

**QTRAN.** QTRAN gives a sufficient condition for value factorisation that satisfies IGM such that

$$\sum_{i \in \mathcal{N}} Q_i(\mathbf{s}, a_i) - Q^\pi(\mathbf{s}, \mathbf{a}) + V^\pi(\mathbf{s}) = \begin{cases} 0 & \mathbf{a} = \bar{\mathbf{a}}, \\ \geq 0 & \mathbf{a} \neq \bar{\mathbf{a}}, \end{cases} \tag{18}$$

wherein

$$V^\pi(\mathbf{s}) = \max_{\mathbf{a}} Q^\pi(\mathbf{s}, \mathbf{a}) - \sum_{i \in \mathcal{N}} Q_i(\mathbf{s}, \bar{a}_i).$$

In Eq.18, $\mathbf{a} = \times_{i \in \mathcal{N}} a_i$; and $\bar{\mathbf{a}} = \times_{i \in \mathcal{N}} \bar{a}_i$ where $\bar{a}_i = \arg\max_{a_i} Q_i(\mathbf{s}, a_i)$ because of IGM. Additionally, Son et al. [10] showed that the above condition also holds for affine transformation on $Q_i, \forall i \in \mathcal{N}$ such that $w_i Q_i + b_i$. For this reason, an additional transformed global Q-value such that $Q^{\pi'}(\mathbf{s}, \mathbf{a}) = \sum_{i \in \mathcal{N}} Q_i(\mathbf{s}, a_i)$ by setting $w_i = 1$ and $\sum_{i \in \mathcal{N}} b_i = 0$ is used to represent the value factorisation. It is forced to fit the above condition with a learned global Q-value $Q^\pi(\mathbf{s}, \mathbf{a})$ and $V^\pi(\mathbf{s})$. Son et al. [10] argued that finding the factorisation of $Q^{\pi'}(\mathbf{s}, \mathbf{a})$ is equivalent to finding $[Q_i]_{i \in \mathcal{N}}$ to satisfy IGM. Therefore, a value factorisation for obtaining decentralised Q-values that satisfies IGM is found.

### D.2 Interpretation of Definitions in Markov Convex Game

#### D.2.1 Condition of Markov Convex Game

Eq.1 implies a fact existing in most real-life scenarios that a larger coalition results in the greater payoff distributions (see Remark 3) and therefore the greater optimal global value in cooperation, which directly increases the agents' incentives for joining the grand coalition. This can be seen as an insight into the global reward game with value factorisation. This interpretation for the dynamic scenario in this paper is consistent with the static scenario given by [44], also known as the snowball effect.

**Remark 3.** *Suppose there are two coalitions $\mathcal{T}, \mathcal{S}$ such that $\mathcal{T} \subset \mathcal{S} \subset \mathcal{N}$ and an agent $i \in \mathcal{N} \backslash \mathcal{S}$. For convenience, we denote $\mathcal{C}_1 = \mathcal{T} \cup \{i\}$ and $\mathcal{C}_2 = \mathcal{S}$, and thus $\mathcal{C}_\cap = \mathcal{C}_1 \cap \mathcal{C}_2 = (\mathcal{T} \cup \{i\}) \cap \mathcal{S} = \mathcal{T}$ and $\mathcal{C}_\cup = \mathcal{C}_1 \cup \mathcal{C}_2 = (\mathcal{T} \cup \{i\}) \cup \mathcal{S} = \mathcal{S} \cup \{i\}$. By Eq.1, we can write the following inequalities such that*

$$\max_{\pi_{\mathcal{S} \cup \{i\}}} V^{\pi_{\mathcal{S} \cup \{i\}}}(\mathbf{s}) - \max_{\pi_{\mathcal{S}}} V^{\pi_{\mathcal{S}}}(\mathbf{s}) = \max_{\pi_{\mathcal{C}_\cup}} V^{\pi_{\mathcal{C}_\cup}}(\mathbf{s}) - \max_{\pi_{\mathcal{C}_2}} V^{\pi_{\mathcal{C}_2}}(\mathbf{s})$$

$$\geq \max_{\pi_{\mathcal{C}_1}} V^{\pi_{\mathcal{C}_1}}(\mathbf{s}) - \max_{\pi_{\mathcal{C}_\cap}} V^{\pi_{\mathcal{C}_\cap}}(\mathbf{s}) \tag{19}$$

$$= \max_{\pi_{\mathcal{T} \cup \{i\}}} V^{\pi_{\mathcal{T} \cup \{i\}}}(\mathbf{s}) - \max_{\pi_{\mathcal{T}}} V^{\pi_{\mathcal{T}}}(\mathbf{s}).$$

*It is intuitive to see that each agent can gain more payoffs if the size of the coalition grows.*

### D.2.2 Insight into Markov Core

In Eq.2, $\left(\max_{\pi_i} x_i(\mathbf{s})\right)_{i\in\mathcal{N}}$ indicates the payoff distribution scheme for the grand coalition. $\max_{\pi_\mathcal{C}} x(\mathbf{s}|\mathcal{C}) = \sum_{i\in\mathcal{C}} \max_{\pi_i} x_i(\mathbf{s})$ indicates the sum of payoff distributions (for the grand coalition) of the agents who is under evaluation within coalition $\mathcal{C}$. By Remark 4 and 5, it is obvious that Eq.2 indicates that the optimal global value obtained by the payoff distribution scheme in the Markov core (under the grand coalition) is no less than that they can achieve with other coalition structures, which is called the maximal social welfare in the prior work [13]. It can be regarded as an intuitive interpretation of Markov core (under the grand coalition).

**Remark 4.** *Suppose that a coalition structure is written as $\mathcal{CS} = \{\mathcal{C}_1, \mathcal{C}_2, ..., \mathcal{C}_n\}$, where $\bigcup_{k=1}^{n} \mathcal{C}_k = \mathcal{N}$ and each $\mathcal{C}_k$ is mutually exclusive (i.e., $\mathcal{C}_m \cap \mathcal{C}_n = \varnothing, if\ m \neq n$), the optimal global value with respect to $\mathcal{CS}$ is represented as $\max_\pi V^\pi(\mathbf{s}) = \sum_{k=1}^{n} \max_{\pi_{\mathcal{C}_k}} V^{\pi_{\mathcal{C}_k}}(\mathbf{s})$.*

**Remark 5.** *Suppose that the condition of Markov core holds for the grand coalition (i.e., $\mathcal{N}$) with some payoff distribution scheme $\left(\max_{\pi_i} x_i(\mathbf{s})\right)_{i\in\mathcal{N}}$. For an arbitrary coalition structure $\mathcal{CS} = \{\mathcal{C}_1, \mathcal{C}_2, ..., \mathcal{C}_n\}$ other than $\{\mathcal{N}\}$, where $\bigcup_{k=1}^{n} \mathcal{C}_k = \mathcal{N}$ and each $\mathcal{C}_k$ is mutually exclusive, we can write down the equation such that*

$$\max_{\pi_{\mathcal{C}_k}} x(\mathbf{s}|\mathcal{C}_k) \geq \max_{\pi_{\mathcal{C}_k}} V^{\pi_{\mathcal{C}_k}}(\mathbf{s}), \quad \forall \mathcal{C}_k \in \mathcal{CS}. \tag{20}$$

*If we sum up Eq.20 for all coalitions in $\mathcal{CS}$, we can get the following equation such that*

$$\sum_{\mathcal{C}_k \in \mathcal{CS}} \max_{\pi_{\mathcal{C}_k}} x(\mathbf{s}|\mathcal{C}_k) \geq \sum_{\mathcal{C}_k \in \mathcal{CS}} \max_{\pi_{\mathcal{C}_k}} V^{\pi_{\mathcal{C}_k}}. \tag{21}$$

*Recall that $\max_{\pi_{\mathcal{C}_k}} x(\mathbf{s}|\mathcal{C}_k) = \sum_{j\in\mathcal{C}_k} \max_{\pi_i} x_i(\mathbf{s})$. The LHS of Eq.21 can be written as follows:*

$$\sum_{\mathcal{C}_k \in \mathcal{CS}} \max_{\pi_{\mathcal{C}_k}} x(\mathbf{s}|\mathcal{C}_k) = \sum_{\mathcal{C}_k \in \mathcal{CS}} \sum_{j\in\mathcal{C}_k} \max_{\pi_j} x_j(\mathbf{s}) = \sum_{j\in\mathcal{N}} \max_{\pi_j} x_j(\mathbf{s}) = \max_\pi \hat{V}^\pi(\mathbf{s}), \tag{22}$$

*wherein $\max_\pi \hat{V}^\pi(\mathbf{s})$ is denoted as the optimal global value obtained by the payoff distribution scheme in the Markov core. By the result in Remark 4, the RHS of Eq.21 can be written as follows:*

$$\sum_{\mathcal{C}_k \in \mathcal{CS}} \max_{\pi_{\mathcal{C}_k}} V^{\pi_{\mathcal{C}_k}} = \max_\pi V^\pi(\mathbf{s}), \tag{23}$$

*where $\max_\pi V^\pi(\mathbf{s})$ is the optimal global value obtained by an arbitrary coalition structure other than $\{\mathcal{N}\}$. By inserting Eq.22 and 23 into Eq.21, we can get that*

$$\max_\pi \hat{V}^\pi(\mathbf{s}) \geq \max_\pi V^\pi(\mathbf{s}).$$

*Therefore, we have shown that the solution in the Markov core under the grand coalition is equivalent to the optimal global value.*

## E   Complete Mathematical Proofs

### E.1   Assumptions

**Assumption 1.** *In this paper, we consider a finite Markov convex game, wherein both the state space and the joint action space are finite.*

**Assumption 2.** *For the ease of analysis, in this paper we assume that each agent's policy will not be affected by the coalition formation. In other words, each agent's policy is regarded as its inherent feature, invariant throughout the interaction with other agents (e.g. joining a coalition).*

**Assumption 3.** *Any coalition policy can be factorised to a permutation of decentralised (i.e. disjoint) policies, i.e., $\pi_\mathcal{C} = \times_{i\in\mathcal{C}} \pi_i$, where $\pi_i$ is agent $i$'s policy. Each $\pi_\mathcal{C}$ uniquely corresponds to a $V^{\pi_\mathcal{C}}(\mathbf{s})$ as a characteristic function (i.e. a set-valued function).*

**Assumption 4.** *If an agent $i$ is a dummy for an arbitrary state $\mathbf{s} \in \mathcal{S}$, it will not provide any contribution to any coalition $\mathcal{C}_i \subseteq \mathcal{N}\backslash\{i\}$ such that $V^{\pi_\mathcal{C}}(\mathbf{s}) = V^{\pi_{\mathcal{C}\cup\{i\}}}(\mathbf{s})$. Additionally, no members in coalition $\mathcal{C}_i$ will react in different manners after agent $i$ joins.*

**Assumption 5.** *If agents $i$ and $j$ are symmetric for an arbitrary state $\mathbf{s} \in \mathcal{S}$, $V^{\pi_{\mathcal{C}\cup\{i\}}}(\mathbf{s}) = V^{\pi_{\mathcal{C}\cup\{j\}}}(\mathbf{s})$ to any coalitions $\mathcal{C} \subseteq \mathcal{N}\backslash\{i,j\}$. Literally, the contributions of $i$ and $j$ are equal to any coalition $\mathcal{C}$.*

**Assumption 6.** *For any agent $i \in \mathcal{N}$ and any $\mathbf{s} \in \mathcal{S}$, its optimal Markov Shapley value denoted as $\max_{\pi_i} V_i^\phi(\mathbf{s})$ satisfies the following equation such that*

$$\max_{\pi_i} V_i^\phi(\mathbf{s}) = \sum_{\mathcal{C}_i \subseteq \mathcal{N} \backslash \{i\}} \frac{|\mathcal{C}_i|!(|\mathcal{N}| - |\mathcal{C}_i| - 1)!}{|\mathcal{N}|!} \cdot \max_{\pi_i} \Phi_i(\mathbf{s}|\mathcal{C}_i),$$

*where $\pi_i$ is agent $i$'s policy.*

Assumption 1 is the common assumption in the Markov decision process for the ease of analysis. Assumption 2 is a technical assumption for the ease of analysis. Assumption 3 is natural to hold given the chain rule in probability theory, the independence of each agent's policy and the definition of value function in reinforcement learning. Assumption 4 and 5 directly inherit the definitions from cooperative game theory [14]. Assumption 6 inherits the definition from Shapley value [19] with extra consideration of agent $i$'s policy, an underlying condition of which is that the maximizer (i.e., $\pi_i$) of each $\Phi_i(\mathbf{s} \mid \mathcal{C}_i) \in \{\Phi_i(\mathbf{s}|\mathcal{C}_i)|\mathcal{C}_i \subseteq \mathcal{N} \backslash \{i\}\}$ needs to be identical, for any $\mathbf{s} \in \mathcal{S}$. In other words, it implies that different permutations correspond to different long-term rewards probably encoding some unexpected events (i.e., each permutation maps to a marginal contribution of agent $i$), but with the same optimal policy as solutions, which is a sufficient condition for Assumption 2. Thereby, learning through Markov Shapley value is primarily for fair credit assignments, with no changes to each agent's optimal policy. We would argue for the existence of this condition by Example 1.

**Example 1.** *Suppose that there are two agents in total (i.e., $|\mathcal{N}| = 2$), and we consider an arbitrary agent $i$ belonging to $\mathcal{N}$ whose action set is defined as $\mathcal{A}_i = \{0, 0.15, 0.25\}$. Therefore, there are only two intermediate coalitions for agent $i$ to join and therefore two marginal contributions. To ease life, we only discuss a two-stage scenario and the result can be naturally extended to long-horizon scenarios. Agent $i$'s policy can be expressed as a sequence of actions such that $\pi_i = \langle a_i^0, a_i^1 \rangle$. The set of marginal contributions of agent $i$ is supposed to be $\{\Phi_i(\mathbf{s}|\{-i\}) := -(a_i^0 + a_i^1 - 0.5)^2 + 1 + \|\mathbf{s}\|_2, \Phi_i(\mathbf{s}|\varnothing) := \sin(a_i^0 + a_i^1) + \|\mathbf{s}\|_2\}$. Since $V_i^\phi(\mathbf{s}) = \frac{1}{2}(\Phi_i(\mathbf{s}|\{-i\}) + \Phi_i(\mathbf{s}|\varnothing))$, it is easy to observe that Assumption 6 holds.*

### E.2 Mathematical Proofs of The Marginal Contribution

**Proposition 4.** *$\forall \mathcal{C}_i \subseteq \mathcal{N}$ and $\forall \mathbf{s} \in \mathcal{S}$, Eq.1 is satisfied if and only if $\max_{\pi_i} \Phi_i(\mathbf{s}|\mathcal{C}_i) \geq 0$.*

*Proof.* $\forall \mathcal{C}_i \subseteq \mathcal{N}$ and $\forall \mathbf{s} \in \mathcal{S}$, given that Eq.1 is satisfied, with the fact that $\mathcal{C}_i \cap \{i\} = \varnothing$ we can get the equation such that

$$\max_{\pi_{\mathcal{C}_i \cup \{i\}}} V^{\pi_{\mathcal{C}_i \cup \{i\}}}(\mathbf{s}) \geq \max_{\pi_{\mathcal{C}_i}} V^{\pi_{\mathcal{C}_i}}(\mathbf{s}) + \max_{\pi_i} V^{\pi_i}(\mathbf{s}). \tag{24}$$

Since $\max_{\pi_i} V^{\pi_i}(\mathbf{s}) \geq 0$ by the definition in Markov convex game, we can easily get the equation such that

$$\max_{\pi_{\mathcal{C}_i \cup \{i\}}} V^{\pi_{\mathcal{C}_i \cup \{i\}}}(\mathbf{s}) - \max_{\pi_{\mathcal{C}_i}} V^{\pi_{\mathcal{C}_i}}(\mathbf{s}) \geq 0. \tag{25}$$

Therefore, we can get the equation such that

$$\max_{\pi_i} \Phi_i(\mathbf{s}|\mathcal{C}_i) \geq 0. \tag{26}$$

With the same conditions, the reverse direction of proof apparently holds by going through from Eq.26 to 24. By Definition 2, Eq.26 determines the range of Markov Shapley value, which is consistent with the range of the coalition value defined in Section 2. $\square$

**Proposition 5.** *In Markov convex game with the grand coalition, marginal contribution satisfies the efficiency property: $\max_{\pi} V^\pi(\mathbf{s}) = \sum_{i \in \mathcal{N}} \max_{\pi_i} \Phi_i(\mathbf{s}|\mathcal{C}_i)$.*

*Proof.* For any $\mathcal{C}_i \subseteq \mathcal{N} \backslash \{i\}$ and $i \in \mathcal{N}$, according to Eq.3 we can get the equation such that

$$\max_{\pi_i} \Phi_i(\mathbf{s}|\mathcal{C}_i) = \max_{\pi_{\mathcal{C}_i \cup \{i\}}} V^{\pi_{\mathcal{C}_i \cup \{i\}}}(\mathbf{s}) - \max_{\pi_{\mathcal{C}_i}} V^{\pi_{\mathcal{C}_i}}(\mathbf{s}), \tag{27}$$

where $\max_{\pi_{\mathcal{C}_i \cup \{i\}}} V^{\pi_{\mathcal{C}_i}}(\mathbf{s}) = \max_{\pi_{\mathcal{C}_i}} V^{\pi_{\mathcal{C}_i}}(\mathbf{s})$, since the decision of agent $i$ will not affect the value of $\mathcal{C}_i$ (i.e., the coalition excluding agent $i$). Given the definition that $V^{\pi_\varnothing}(\mathbf{s}) = 0$ and the result from Eq.27, by Assumption 3 we can get the equations such that

$$
\begin{aligned}
&\max_{\pi} V^{\pi}(\mathbf{s}) \\
&= \max_{\pi_{\{j_1\}}} V^{\pi_{\{j_1\}}}(\mathbf{s}) - \max_{\pi_\varnothing} V^{\pi_\varnothing}(\mathbf{s}) \\
&+ \max_{\pi_{\{j_1, j_2\}}} V^{\pi_{\{j_1\}}}(\mathbf{s}) - \max_{\pi_{\{j_1\}}} V^{\pi_{\{j_1\}}}(\mathbf{s}) \\
&+ \qquad\qquad \vdots \\
&+ \max_{\pi} V^{\pi}(\mathbf{s}) - \max_{\pi_{\mathcal{N} \setminus \{j_n\}}} V^{\pi_{\mathcal{N} \setminus \{j_n\}}}(\mathbf{s}) \\
&= \sum_{i \in \mathcal{N}} \max_{\pi_i} \Phi_i(\mathbf{s}|\mathcal{C}_i).
\end{aligned}
\tag{28}
$$

$\square$

**Lemma 1.** *The optimal marginal contribution is a solution in the Markov core under Markov convex game with the grand coalition.*

*Proof.* The complete proof is as follows.

Firstly, if we would like to prove that the optimal marginal contribution is a payoff distribution scheme in the Markov core (with the grand coalition), we just need to prove that for any intermediate coalition $\mathcal{C} \subseteq \mathcal{N}$, the following condition is satisfied such that

$$
\max_{\pi_{\mathcal{C}}} \Phi(\mathbf{s}|\mathcal{C}) \geq \max_{\pi_{\mathcal{C}}} V^{\pi_{\mathcal{C}}}(\mathbf{s}), \ \forall \mathbf{s} \in \mathcal{S},
\tag{29}
$$

where $\max_{\pi_{\mathcal{C}}} \Phi(\mathbf{s}|\mathcal{C}) = \sum_{i \in \mathcal{C}} \max_{\pi_i} \Phi_i(\mathbf{s}|\mathcal{C}_i)$.

Suppose for the sake of contradiction that we have $\max_{\pi_{\mathcal{C}}} \Phi(\mathbf{s}|\mathcal{C}) < \max_{\pi_{\mathcal{C}}} V^{\pi_{\mathcal{C}}}(\mathbf{s})$ for some $\mathbf{s} \in \mathcal{S}$ and some coalition $\mathcal{C} = \{j_1, j_2, ..., j_{|\mathcal{C}|}\} \subseteq \mathcal{N}$, where $j_n \in \mathcal{C}$ and $n \in \{1, 2, ..., |\mathcal{C}|\}$. We can assume without the loss of generality that the coalition $\mathcal{C}$ is generated by the permutation $\langle j_1, j_2, ..., j_{|\mathcal{C}|} \rangle$, i.e., the agents joins in $\mathcal{C}$ following the order $j_1, j_2, ..., j_{|\mathcal{C}|}$. Now, for each $n \in \{1, 2, ..., |\mathcal{C}|\}$, we have $\{j_1, j_2, ..., j_{n-1}\} \subseteq \{1, 2, ..., j_n - 1\}$. Following Eq.1, we can write out the inequality as follows:

$$
\max_{\pi_{\mathcal{C}_\cup^n}} V^{\pi_{\mathcal{C}_\cup^n}}(\mathbf{s}) + \max_{\pi_{\mathcal{C}_\cap^n}} V^{\pi_{\mathcal{C}_\cap^n}}(\mathbf{s}) \geq \max_{\pi_{\mathcal{C}_m^n}} V^{\pi_{\mathcal{C}_m^n}}(\mathbf{s}) + \max_{\pi_{\mathcal{C}_k^n}} V^{\pi_{\mathcal{C}_k^n}}(\mathbf{s}),
$$
$$
\mathcal{C}_k^n = \{1, 2, ..., j_n - 1\},
$$
$$
\mathcal{C}_m^n = \{j_1, j_2, ..., j_n\},
\tag{30}
$$
$$
\mathcal{C}_\cap^n = \mathcal{C}_m^n \cap \mathcal{C}_k^n = \{j_1, j_2, ..., j_{n-1}\},
$$
$$
\mathcal{C}_\cup^n = \mathcal{C}_m^n \cup \mathcal{C}_k^n = \{1, 2, ..., j_n\}.
$$

Next, we rearrange Eq.30 and the following inequality is obtained such that

$$
\max_{\pi_{\mathcal{C}_\cup^n}} V^{\pi_{\mathcal{C}_\cup^n}}(\mathbf{s}) - \max_{\pi_{\mathcal{C}_k^n}} V^{\pi_{\mathcal{C}_k^n}}(\mathbf{s}) \geq \max_{\pi_{\mathcal{C}_m^n}} V^{\pi_{\mathcal{C}_m^n}}(\mathbf{s}) - \max_{\pi_{\mathcal{C}_\cap^n}} V^{\pi_{\mathcal{C}_\cap^n}}(\mathbf{s}),
\tag{31}
$$

Since we can express $\max_{\pi_{\mathcal{C}}} V^{\pi_{\mathcal{C}}}(\mathbf{s})$ as follows:

$$
\begin{aligned}
\max_{\pi_{\mathcal{C}}} V^{\pi_{\mathcal{C}}}(\mathbf{s}) &= \max_{\pi_{j_1}} V^{\pi_{j_1}}(\mathbf{s}) - \max_{\pi_\varnothing} V^{\pi_\varnothing}(\mathbf{s}) \\
&+ \max_{\pi_{\{j_1, j_2\}}} V^{\pi_{\{j_1, j_2\}}}(\mathbf{s}) - \max_{\pi_{j_1}} V^{\pi_{j_1}}(\mathbf{s}) \\
&+ \qquad\qquad \vdots \\
&+ \max_{\pi_{\mathcal{C}}} V^{\pi_{\mathcal{C}}}(\mathbf{s}) - \max_{\pi_{\mathcal{C} \setminus \{j_n\}}} V^{\pi_{\mathcal{C} \setminus \{j_n\}}}(\mathbf{s}).
\end{aligned}
\tag{32}
$$

By Definition 1 we can obviously get the following equations such that

$$
\Phi_i(\mathbf{s}|\mathcal{C}_i) = \Phi_i(\mathbf{s}|\mathcal{C}_k^n) = \max_{\pi_{\mathcal{C}_\cup^n}} V^{\pi_{\mathcal{C}_\cup^n}}(\mathbf{s}) - \max_{\pi_{\mathcal{C}_k^n}} V^{\pi_{\mathcal{C}_k^n}}(\mathbf{s}).
\tag{33}
$$

By taking the maximum operator over $\pi_i$ to Eq.33, we can get that

$$\max_{\pi_i} \Phi_i(\mathbf{s}|\mathcal{C}_i) = \max_{\pi_i} \Phi_i(\mathbf{s}|\mathcal{C}_k^n) = \max_{\pi_{\mathcal{C}_\cup^n}} V^{\pi_{\mathcal{C}_\cup^n}}(\mathbf{s}) - \max_{\pi_{\mathcal{C}_k^n}} V^{\pi_{\mathcal{C}_k^n}}(\mathbf{s}). \tag{34}$$

By adding up these inequalities in Eq.31 for all $\mathcal{C} \subseteq \mathcal{N}$ and inserting the results from Eq.32 and 34, we can directly obtain a new inequality such that

$$\sum_{i \in \mathcal{C}} \max_{\pi_i} \Phi_i(\mathbf{s}|\mathcal{C}_i) = \max_{\pi_\mathcal{C}} \Phi(\mathbf{s}|\mathcal{C}) \geq \max_{\pi_\mathcal{C}} V^{\pi_\mathcal{C}}(\mathbf{s}). \tag{35}$$

It is obvious that Eq.35 contradicts the suppose, so we have showed that Eq.29 always holds for any coalition $\mathcal{C} \subseteq \mathcal{N}$. For this reason, we can get the conclusion that marginal contribution is a solution in Markov core of Markov convex game with the grand coalition. $\square$

### E.3 Mathematical Proofs of The Markov Shapley Value

**Proposition 1.** *Agent $i$'s action marginal contribution can be derived as follows:*

$$\Phi_i(\mathbf{s}, a_i|\mathcal{C}_i) = \max_{\mathbf{a}_{\mathcal{C}_i}} Q^{\pi_{\mathcal{C}_i}^*}(\mathbf{s}, \mathbf{a}_{\mathcal{C}_i \cup \{i\}}) - \max_{\mathbf{a}_{\mathcal{C}_i}} Q^{\pi_{\mathcal{C}_i}^*}(\mathbf{s}, \mathbf{a}_{\mathcal{C}_i}). \tag{36}$$

*Proof.* The complete proof is as follows.

We now rewrite $\max_{\pi_{\mathcal{C}_i}} V^{\pi_{\mathcal{C}_i \cup \{i\}}}(\mathbf{s})$ as follows:

$$\max_{\pi_{\mathcal{C}_i}} V^{\pi_{\mathcal{C}_i \cup \{i\}}}(\mathbf{s}) = \max_{\pi_{\mathcal{C}_i}} \sum_{\mathbf{a}_{\mathcal{C}_i \cup \{i\}}} \pi_{\mathcal{C}_i \cup \{i\}}(\mathbf{a}_{\mathcal{C}_i \cup \{i\}}|\mathbf{s}) \, Q^{\pi_{\mathcal{C}_i \cup \{i\}}}(\mathbf{s}, \mathbf{a}_{\mathcal{C}_i \cup \{i\}})$$

$$\left(\text{Since } \pi_{\mathcal{C}_i \cup \{i\}} \text{ is a deterministic joint policy, we can have the following equation.}\right)$$

$$= \max_{\mathbf{a}_{\mathcal{C}_i}} \max_{\pi_{\mathcal{C}_i}} Q^{\pi_{\mathcal{C}_i \cup \{i\}}}(\mathbf{s}, \mathbf{a}_{\mathcal{C}_i \cup \{i\}})$$

$$\left(\text{We write } \max_{\pi_{\mathcal{C}_i}} Q^{\pi_{\mathcal{C}_i \cup \{i\}}}(\mathbf{s}, \mathbf{a}_{\mathcal{C}_i \cup \{i\}}) \text{ as } Q^{\pi_{\mathcal{C}_i}^*}(\mathbf{s}, \mathbf{a}_{\mathcal{C}_i \cup \{i\}})\right)$$

$$= \max_{\mathbf{a}_{\mathcal{C}_i}} Q^{\pi_{\mathcal{C}_i}^*}(\mathbf{s}, \mathbf{a}_{\mathcal{C}_i \cup \{i\}}). \tag{37}$$

Similarly, we rewrite $\max_{\pi_{\mathcal{C}_i}} V^{\pi_{\mathcal{C}_i}}(\mathbf{s})$ as follows:

$$\max_{\pi_{\mathcal{C}_i}} V^{\pi_{\mathcal{C}_i}}(\mathbf{s}) = \max_{\mathbf{a}_{\mathcal{C}_i}} \max_{\pi_{\mathcal{C}_i}} Q^{\pi_{\mathcal{C}_i}}(\mathbf{s}, \mathbf{a}_{\mathcal{C}_i}) = \max_{\mathbf{a}_{\mathcal{C}_i}} Q^{\pi_{\mathcal{C}_i}^*}(\mathbf{s}, \mathbf{a}_{\mathcal{C}_i}). \tag{38}$$

Since $\max_{\pi_{\mathcal{C}_i}} V^{\pi_{\mathcal{C}_i}}(\mathbf{s})$ is irrelevant to $a_i$, by Eq.37 and 38 we can get that

$$\Phi_i(\mathbf{s}, a_i|\mathcal{C}_i) = \max_{\mathbf{a}_{\mathcal{C}_i}} Q^{\pi_{\mathcal{C}_i}^*}(\mathbf{s}, \mathbf{a}_{\mathcal{C}_i \cup \{i\}}) - \max_{\mathbf{a}_{\mathcal{C}_i}} Q^{\pi_{\mathcal{C}_i}^*}(\mathbf{s}, \mathbf{a}_{\mathcal{C}_i}). \tag{39}$$

By Eq.39, we can also get Agent $i$'s optimal action marginal contribution such that

$$\Phi_i^*(\mathbf{s}, a_i|\mathcal{C}_i) = \max_{\pi_i} \Phi_i(\mathbf{s}, a_i|\mathcal{C}_i)$$

$$= \max_{\pi_i} \left\{ \max_{\mathbf{a}_{\mathcal{C}_i}} Q^{\pi_{\mathcal{C}_i}^*}(\mathbf{s}, \mathbf{a}_{\mathcal{C}_i \cup \{i\}}) - \max_{\mathbf{a}_{\mathcal{C}_i}} Q^{\pi_{\mathcal{C}_i}^*}(\mathbf{s}, \mathbf{a}_{\mathcal{C}_i}) \right\}$$

$$= \max_{\pi_i} \left\{ \max_{\mathbf{a}_{\mathcal{C}_i}} \max_{\pi_{\mathcal{C}_i}} Q^{\pi_{\mathcal{C}_i \cup \{i\}}}(\mathbf{s}, \mathbf{a}_{\mathcal{C}_i \cup \{i\}}) - \max_{\mathbf{a}_{\mathcal{C}_i}} \max_{\pi_{\mathcal{C}_i}} Q^{\pi_{\mathcal{C}_i}}(\mathbf{s}, \mathbf{a}_{\mathcal{C}_i}) \right\}$$

$$= \max_{\pi_i} \max_{\mathbf{a}_{\mathcal{C}_i}} \max_{\pi_{\mathcal{C}_i}} Q^{\pi_{\mathcal{C}_i \cup \{i\}}}(\mathbf{s}, \mathbf{a}_{\mathcal{C}_i \cup \{i\}}) - \max_{\mathbf{a}_{\mathcal{C}_i}} \max_{\pi_{\mathcal{C}_i}} Q^{\pi_{\mathcal{C}_i}}(\mathbf{s}, \mathbf{a}_{\mathcal{C}_i})$$

$$= \max_{\mathbf{a}_{\mathcal{C}_i}} \max_{\pi_{\mathcal{C}_i \cup \{i\}}} Q^{\pi_{\mathcal{C}_i \cup \{i\}}}(\mathbf{s}, \mathbf{a}_{\mathcal{C}_i \cup \{i\}}) - \max_{\mathbf{a}_{\mathcal{C}_i}} \max_{\pi_{\mathcal{C}_i}} Q^{\pi_{\mathcal{C}_i}}(\mathbf{s}, \mathbf{a}_{\mathcal{C}_i})$$

$$= \max_{\mathbf{a}_{\mathcal{C}_i}} Q^{\pi_{\mathcal{C}_i \cup \{i\}}^*}(\mathbf{s}, \mathbf{a}_{\mathcal{C}_i \cup \{i\}}) - \max_{\mathbf{a}_{\mathcal{C}_i}} Q^{\pi_{\mathcal{C}_i}^*}(\mathbf{s}, \mathbf{a}_{\mathcal{C}_i}). \tag{40}$$

The proof is completed. $\square$

**Proposition 2.** *Markov Shapley value possesses properties as follows: (i) identifiability of dummy agents: $V_i^\phi(\mathbf{s}) = 0$; (ii) efficiency: $\max_\pi V^\pi(\mathbf{s}) = \sum_{i\in\mathcal{N}} \max_{\pi_i} V_i^\phi(\mathbf{s})$; (iii) reflecting the contribution; and (iv) symmetry.*

*Proof.* The complete proof is as follows.

The marginal contribution is an implementation reflecting an agent's contribution and Markov Shapley value is defined as the weighted average of all marginal contributions. Therefore, this definition can still reflect an agent's contribution to the grand coalition by considering all permutations of agents to form the grand coalition and (iii) holds. We will next prove the (i), followed by (ii) and (iv). For any agent $i \in \mathcal{N}$ and any state $\mathbf{s} \in \mathcal{S}$, its Markov Shapley value denoted as $V_i^\phi(\mathbf{s})$.

**Proof of (i):** Let us define $\Pi(\mathcal{N})$ as the set of all permutations of agents. Suppose that an arbitrary agent $i$ is a dummy agent for an arbitrary state $\mathbf{s} \in \mathcal{S}$. For any permutation $m \in \Pi(\mathcal{N})$ of agents to form the grand coalition, by Assumption 4 we have $\max_{\pi_{\mathcal{C}_i^m}} V^{\pi_{\mathcal{C}_i^m}}(\mathbf{s}) = \max_{\pi_{\mathcal{C}_i^m}} V^{\pi_{\mathcal{C}_i^m \cup \{i\}}}(\mathbf{s})$, thereby $\Phi_i(\mathbf{s}|\mathcal{C}_i^m) = 0$, where $\mathcal{C}_i^m$ denotes the intermediate coalition generated from permutation $m$ that agent $i$ would join. Also, the above analysis is valid for all permutations of agents to form the grand coalition. By Definition 2, it is not difficult to see that the dummy agent's Markov Shapley value will be 0 such that $V_i^\phi(\mathbf{s}) = 0$. The proof of (i) completes.

**Proof of (ii):** The objective is proving that Markov Shapley value satisfies the following equation such that

$$\max_\pi V^\pi(\mathbf{s}) = \sum_{i\in\mathcal{N}} \max_{\pi_i} V_i^\phi(\mathbf{s}), \quad \forall \mathbf{s} \in \mathcal{S}.$$

By the result from Proposition 5 and Assumption 3, for an arbitrary permutation $m \in \Pi(\mathcal{N})$ we can get the equation such that

$$\max_\pi V^\pi(\mathbf{s}) = \sum_{i\in\mathcal{N}} \max_{\pi_i} \Phi_i(\mathbf{s}|\mathcal{C}_i^m), \quad \forall \mathbf{s} \in \mathcal{S},$$

where $\mathcal{C}_i^m$ denotes the intermediate coalition generated from permutation $m$ that agent $i$ would join and $\Phi_i(\mathbf{s}|\mathcal{C}_i^m)$ is the corresponding marginal contribution. If we consider all possible permutations of agents to form the grand coalition and add all these inequalities, we can get the following equation such that

$$\sum_{m\in\Pi(\mathcal{N})} \max_\pi V^\pi(\mathbf{s}) = \sum_{m\in\Pi(\mathcal{N})} \sum_{i\in\mathcal{N}} \max_{\pi_i} \Phi_i(\mathbf{s}|\mathcal{C}_i^m), \quad \forall \mathbf{s} \in \mathcal{S}.$$

By dividing $|\mathcal{N}|!$ on the both sides, we can get that

$$\frac{1}{|\mathcal{N}|!} \sum_{m\in\Pi(\mathcal{N})} \max_\pi V^\pi(\mathbf{s}) = \frac{1}{|\mathcal{N}|!} \sum_{i\in\mathcal{N}} \sum_{m\in\Pi(\mathcal{N})} \max_{\pi_i} \Phi_i(\mathbf{s}|\mathcal{C}_i^m), \quad \forall \mathbf{s} \in \mathcal{S}. \tag{41}$$

Next, to ease life we start from the LHS of Eq.41. We directly get the following equation such that

$$\frac{1}{|\mathcal{N}|!} \sum_{m\in\Pi(\mathcal{N})} \max_\pi V^\pi(\mathbf{s}) = \frac{1}{|\mathcal{N}|!} \cdot |\mathcal{N}|! \cdot \max_\pi V^\pi(\mathbf{s}) = \max_\pi V^\pi(\mathbf{s}). \tag{42}$$

Now, we start processing the RHS of Eq.41. By rearranging it, we can get the equations such that

$$\frac{1}{|\mathcal{N}|!} \sum_{i\in\mathcal{N}} \sum_{m\in\Pi(\mathcal{N})} \max_{\pi_i} \Phi_i(\mathbf{s}|\mathcal{C}_i^m) = \sum_{i\in\mathcal{N}} \frac{1}{|\mathcal{N}|!} \sum_{m\in\Pi(\mathcal{N})} \max_{\pi_i} \Phi_i(\mathbf{s}|\mathcal{C}_i^m)$$

(The identical $\mathcal{C}_i^m$ in different permutations is written as $\mathcal{C}_i$ and we can rearrange the equation as follows.)

$$= \sum_{i\in\mathcal{C}} \frac{1}{|\mathcal{N}|!} \sum_{\mathcal{C}_i\subseteq\mathcal{N}\setminus\{i\}} |\mathcal{C}_i|!(|\mathcal{N}| - |\mathcal{C}_i| - 1)! \cdot \max_{\pi_i} \Phi_i(\mathbf{s}|\mathcal{C}_i)$$

$$= \sum_{i\in\mathcal{N}} \sum_{\mathcal{C}_i\subseteq\mathcal{N}\setminus\{i\}} \frac{|\mathcal{C}_i|!(|\mathcal{N}| - |\mathcal{C}_i| - 1)!}{|\mathcal{N}|!} \cdot \max_{\pi_i} \Phi_i(\mathbf{s}|\mathcal{C}_i). \tag{43}$$

By Assumption 6, we can get the following equations such that

$$\sum_{i \in \mathcal{N}} \sum_{\mathcal{C}_i \subseteq \mathcal{N} \setminus \{i\}} \frac{|\mathcal{C}_i|!(|\mathcal{N}| - |\mathcal{C}_i| - 1)!}{|\mathcal{N}|!} \cdot \max_{\pi_i} \Phi_i(\mathbf{s}|\mathcal{C}_i) = \sum_{i \in \mathcal{N}} \max_{\pi_i} V_i^{\phi}(\mathbf{s}) \qquad (44)$$

Inserting the results from Eq.42 and 44 to Eq.41, we can get the equation such that

$$\max_{\pi} V^{\pi}(\mathbf{s}) = \sum_{i \in \mathcal{N}} \max_{\pi_i} V_i^{\phi}(\mathbf{s}), \quad \forall \mathbf{s} \in \mathcal{S}.$$

Therefore, the proof for (ii) completes.

**Proof of (iv):** We would like to prove that if two agents are symmetric for an arbitrary state $\mathbf{s} \in \mathcal{S}$, then their optimal Markov Shapley values should be equal. As Assumption 5 illustrates, suppose that agents $i$ and $j$ are symmetric for an arbitrary state $\mathbf{s} \in \mathcal{S}$, $V^{\pi_{\mathcal{C} \cup \{i\}}}(\mathbf{s}) = V^{\pi_{\mathcal{C} \cup \{j\}}}(\mathbf{s})$ for any coalitions $\mathcal{C} \subseteq \mathcal{N} \setminus \{i, j\}$. Given an arbitrary permutation $m \in \Pi(\mathcal{N})$, let $m'$ denote the permutation obtained by exchanging $i$ and $j$ such that $\mathcal{C}_i^m = \mathcal{C}_j^{m'}$, $\mathcal{C}_i^{m'} = \mathcal{C}_j^m$ and $\mathcal{C}_l^{m'} = \mathcal{C}_l^m$, $\forall l \neq i, j$. Next, we aim to prove that $\max_{\pi_i} \Phi_i(\mathbf{s}|\mathcal{C}_i^m) = \max_{\pi_j} \Phi_j(\mathbf{s}|\mathcal{C}_j^{m'})$, for the state $\mathbf{s}$.

We first suppose that $i$ precedes $j$ in $m$. Then we have $\mathcal{C}_i^m = \mathcal{C}_j^{m'}$. Setting $\mathcal{C} = \mathcal{C}_i^m = \mathcal{C}_j^{m'}$, for the state $\mathbf{s}$ we can obtain that

$$\max_{\pi_i} \Phi_i(\mathbf{s}|\mathcal{C}_i^m) = \max_{\pi_{\mathcal{C} \cup \{i\}}} V^{\pi_{\mathcal{C} \cup \{i\}}}(\mathbf{s}) - \max_{\pi_{\mathcal{C}}} V^{\pi_{\mathcal{C}}}(\mathbf{s}),$$

$$\max_{\pi_j} \Phi_j(\mathbf{s}|\mathcal{C}_j^{m'}) = \max_{\pi_{\mathcal{C} \cup \{j\}}} V^{\pi_{\mathcal{C} \cup \{j\}}}(\mathbf{s}) - \max_{\pi_{\mathcal{C}}} V^{\pi_{\mathcal{C}}}(\mathbf{s}).$$

By symmetry, we have $V^{\pi_{\mathcal{C} \cup \{i\}}}(\mathbf{s}) = V^{\pi_{\mathcal{C} \cup \{j\}}}(\mathbf{s})$, which directly implies that $\max_{\pi_i} \Phi_i(\mathbf{s}|\mathcal{C}_i^m) = \max_{\pi_j} \Phi_j(\mathbf{s}|\mathcal{C}_j^{m'})$.

Second, we suppose that $j$ precedes $i$ in $m$. Setting $\mathcal{C} = \mathcal{C}_i^m \setminus \{j\}$, for the state $\mathbf{s}$ we have

$$\max_{\pi_i} \Phi_i(\mathbf{s}|\mathcal{C}_i^m) = \max_{\pi_{\mathcal{C} \cup \{j\} \cup \{i\}}} V^{\pi_{\mathcal{C} \cup \{j\} \cup \{i\}}}(\mathbf{s}) - \max_{\pi_{\mathcal{C} \cup \{j\}}} V^{\pi_{\mathcal{C} \cup \{j\}}}(\mathbf{s}),$$

$$\max_{\pi_j} \Phi_j(\mathbf{s}|\mathcal{C}_j^{m'}) = \max_{\pi_{\mathcal{C} \cup \{j\} \cup \{i\}}} V^{\pi_{\mathcal{C} \cup \{j\} \cup \{i\}}}(\mathbf{s}) - \max_{\pi_{\mathcal{C} \cup \{i\}}} V^{\pi_{\mathcal{C} \cup \{i\}}}(\mathbf{s}).$$

Since $\mathcal{C} \subseteq \mathcal{N} \setminus \{i, j\}$, by symmetry we have $V^{\pi_{\mathcal{C} \cup \{j\}}}(\mathbf{s}) = V^{\pi_{\mathcal{C} \cup \{i\}}}(\mathbf{s})$ and thus $\max_{\pi_i} \Phi_i(\mathbf{s}|\mathcal{C}_i^m) = \max_{\pi_j} \Phi_j(\mathbf{s}|\mathcal{C}_j^{m'})$. Therefore, we have proved that $\max_{\pi_i} \Phi_i(\mathbf{s}|\mathcal{C}_i^m) = \max_{\pi_j} \Phi_j(\mathbf{s}|\mathcal{C}_j^{m'})$ for any $m \in \Pi(\mathcal{N})$. It is not difficult to observe that $m \mapsto m'$ is a one-to-one mapping, so $\Pi(\mathcal{N}) = \{m' \mid m \in \Pi(\mathcal{N})\}$.

By Assumption 6, for an arbitrary state $\mathbf{s} \in \mathcal{S}$ wherein agents are symmetric, we can directly have

$$\begin{aligned}
\max_{\pi_i} V_i^{\phi}(\mathbf{s}) &= \sum_{\mathcal{C}_i \subseteq \mathcal{N} \setminus \{i\}} \frac{|\mathcal{C}_i|!(|\mathcal{N}| - |\mathcal{C}_i| - 1)!}{|\mathcal{N}|!} \cdot \max_{\pi_i} \Phi_i(\mathbf{s}|\mathcal{C}_i) \\
&= \frac{1}{|\mathcal{N}|!} \sum_{m \in \Pi(\mathcal{N})} \max_{\pi_i} \Phi_i(\mathbf{s}|\mathcal{C}_i^m) \\
&= \frac{1}{|\mathcal{N}|!} \sum_{m' \in \Pi(\mathcal{N})} \max_{\pi_j} \Phi_j(\mathbf{s}|\mathcal{C}_j^{m'}) \\
&= \sum_{\mathcal{C}_j \subseteq \mathcal{N} \setminus \{j\}} \frac{|\mathcal{C}_j|!(|\mathcal{N}| - |\mathcal{C}_j| - 1)!}{|\mathcal{N}|!} \cdot \max_{\pi_j} \Phi_j(\mathbf{s}|\mathcal{C}_j) \\
&= \max_{\pi_j} V_j^{\phi}(\mathbf{s}).
\end{aligned}$$

The proof of (iv) completes. $\qquad\qquad\qquad\qquad\qquad\qquad\qquad\qquad\qquad\qquad\qquad\quad$ $\square$

### E.4 Mathematical Proofs and Derivations for Shapley Q-Learning

#### E.4.1 Derivation of Shapley-Bellman optimality equation.

First, according to Bellman's principle of optimality [15, 16] we can write out Bellman optimality equation for the optimal global Q-value such that

$$Q^{\pi^*}(\mathbf{s}, \mathbf{a}) = \sum_{\mathbf{s}'} Pr(\mathbf{s}'|\mathbf{s}, \mathbf{a}) \left[ R + \gamma \max_{\mathbf{a}} Q^{\pi^*}(\mathbf{s}', \mathbf{a}) \right]. \tag{45}$$

For convenience, we only consider the finite state space and action space here. By the efficiency property (i.e. (ii) in Proposition 2), we can get the approximation of the optimal global Q-value w.r.t. optimal actions such that

$$\max_{\mathbf{a}} Q^{\pi^*}(\mathbf{s}', \mathbf{a}) = \sum_{i \in \mathcal{N}} \max_{a_i} Q_i^{\phi^*}(\mathbf{s}', a_i). \tag{46}$$

Suppose that for all $\mathbf{s} \in \mathcal{S}$ and $a_i \in \mathcal{A}_i$, for each agent $i$ there exists bounded $w_i(\mathbf{s}, a_i) > 0$ and $b_i(\mathbf{s}) \geq 0$ that can project $Q^{\pi^*}(\mathbf{s}, \mathbf{a})$ onto the space of $Q_i^{\phi^*}(\mathbf{s}, a_i)$ such that

$$Q_i^{\phi^*}(\mathbf{s}, a_i) = w_i(\mathbf{s}, a_i) Q^{\pi^*}(\mathbf{s}, \mathbf{a}) - b_i(\mathbf{s}). \tag{47}$$

If we denote $\mathbf{w}(\mathbf{s}, \mathbf{a}) = [w_i(\mathbf{s}, a_i)]^\top \in \mathbb{R}_{>0}^{|\mathcal{N}|}$, $\mathbf{b}(\mathbf{s}) = [b_i(\mathbf{s})]^\top \in \mathbb{R}_{\geq 0}^{|\mathcal{N}|}$ and $\mathbf{Q}^{\phi^*}(\mathbf{s}, \mathbf{a}) = [Q_i^{\phi^*}(\mathbf{s}, a_i)]^\top \in \mathbb{R}_{\geq 0}^{|\mathcal{N}|}$, given Eq.47 we can write that

$$\mathbf{Q}^{\phi^*}(\mathbf{s}, \mathbf{a}) = \mathbf{w}(\mathbf{s}, \mathbf{a}) Q^{\pi^*}(\mathbf{s}, \mathbf{a}) - \mathbf{b}(\mathbf{s}). \tag{48}$$

Besides, we suppose that $\sum_{i \in \mathcal{N}} w_i(\mathbf{s}, a_i)^{-1} b_i(\mathbf{s}) = 0$.

Combined with Eq.46 and 48, we can rewrite Eq.45 to the equation as follows:

$$\mathbf{Q}^{\phi^*}(\mathbf{s}, \mathbf{a}) = \mathbf{w}(\mathbf{s}, \mathbf{a}) \sum_{\mathbf{s}'} Pr(\mathbf{s}'|\mathbf{s}, \mathbf{a}) \left[ R + \gamma \sum_{i \in \mathcal{N}} \max_{a_i} Q_i^{\phi^*}(\mathbf{s}', a_i) \right] - \mathbf{b}(\mathbf{s}). \tag{49}$$

From Eq.47, we know that $w_i(\mathbf{s}, a_i) > 0$. Therefore, we can rewrite Eq.47 to the following equation such that

$$w_i(\mathbf{s}, a_i)^{-1} \left( Q_i^{\phi^*}(\mathbf{s}, a_i) + b_i(\mathbf{s}) \right) = Q^{\pi^*}(\mathbf{s}, \mathbf{a}). \tag{50}$$

If we sum up Eq.50 for all agents, we can obtain that

$$\sum_{i \in \mathcal{N}} w_i(\mathbf{s}, a_i)^{-1} \left( Q_i^{\phi^*}(\mathbf{s}, a_i) + b_i(\mathbf{s}) \right) = |\mathcal{N}| Q^{\pi^*}(\mathbf{s}, \mathbf{a}). \tag{51}$$

Since $\sum_{i \in \mathcal{N}} w_i(\mathbf{s}, a_i)^{-1} b_i(\mathbf{s}) = 0$, we can get the following equation such that

$$\sum_{i \in \mathcal{N}} \frac{1}{|\mathcal{N}| w_i(\mathbf{s}, a_i)} Q_i^{\phi^*}(\mathbf{s}, a_i) = Q^{\pi^*}(\mathbf{s}, \mathbf{a}). \tag{52}$$

Inserting Eq.46 into Eq.52, we can get the following equation such that

$$\max_{\mathbf{a}} \sum_{i \in \mathcal{N}} \frac{1}{|\mathcal{N}| w_i(\mathbf{s}, a_i)} Q_i^{\phi^*}(\mathbf{s}, a_i) = \sum_{i \in \mathcal{N}} \max_{a_i} Q_i^{\phi^*}(\mathbf{s}, a_i). \tag{53}$$

Since $\mathbf{a} = \times_{i \in \mathcal{N}} a_i$, we can get that

$$\sum_{i \in \mathcal{N}} \max_{a_i} \frac{1}{|\mathcal{N}| w_i(\mathbf{s}, a_i)} Q_i^{\phi^*}(\mathbf{s}, a_i) = \sum_{i \in \mathcal{N}} \max_{a_i} Q_i^{\phi^*}(\mathbf{s}, a_i). \tag{54}$$

It is apparent that $\forall \mathbf{s} \in \mathcal{S}$ and $a_i^* = \arg\max_{a_i} Q_i^{\phi^*}(\mathbf{s}, a_i)$, we have a solution $w_i(\mathbf{s}, a_i^*) = 1/|\mathcal{N}|$. [7]

---

[7] Note that it exists other solutions rather than the one that we deduce between $\max_{a_i} \frac{1}{|\mathcal{N}| w_i(\mathbf{s}, a_i)} Q_i^{\phi^*}(\mathbf{s}, a_i)$ and $\max_{a_i} Q_i^{\phi^*}(\mathbf{s}, a_i)$. Nevertheless, the result obtained in this paper is the one that exactly matches and explains the finding in the previous works [20]. As for the reason why the solution is the most likely to be achieved in empirical results is deserved to be studied in the future work.

### E.4.2 Proof of Theorem 1

**Lemma 2** ( Dales et al. [45] ). *A set of real matrices $\mathcal{M}$ with a sub-multiplicative norm is a Banach Algebra and a non-empty complete metric space where the metric is induced by the sub-multiplicative norm. A sub-multiplicative norm $\|\cdot\|$ is a norm satisfying the following inequality such that*

$$\forall \mathbf{A}, \mathbf{B} \in \mathcal{M} : \|\mathbf{AB}\| \leq \|\mathbf{A}\|\,\|\mathbf{B}\|.$$

**Lemma 3.** *For a set of real matrices $\mathcal{M}$, given an arbitrary matrix $\mathbf{A} = [a_{ij}] \in \mathbb{R}^{m \times n}$, $\|\mathbf{A}\|_1 = \max_{1 \leq j \leq n} \sum_{1 \leq i \leq m} |a_{ij}|$ is a sub-multiplicative norm.*

*Proof.* The complete proof is as follows.

First, we select two arbitrary matrices belonging to $\mathcal{M}$, i.e. $\mathbf{A} = [a_{ik}] \in \mathbb{R}^{m \times r}$ and $\mathbf{B} = [b_{kj}] \in \mathbb{R}^{r \times n}$. Then, we start proving that $\|\cdot\|_1$ is a sub-multiplicative norm as follows:

$$
\begin{aligned}
\|\mathbf{AB}\|_1 &= \left\| \left[ \sum_{1 \leq k \leq r} a_{ik} b_{kj} \right] \right\|_1 \\
&= \max_{1 \leq j \leq n} \sum_{1 \leq i \leq m} \left| \sum_{1 \leq k \leq r} a_{ik} b_{kj} \right|
\end{aligned}
$$

(By triangle inequality, we can obtain the following inequality.)

$$
\begin{aligned}
&\leq \max_{1 \leq j \leq n} \sum_{1 \leq i \leq m} \sum_{1 \leq k \leq r} |a_{ik} b_{kj}| \\
&= \max_{1 \leq j \leq n} \sum_{1 \leq i \leq m} \sum_{1 \leq k \leq r} |a_{ik}|\,|b_{kj}| \\
&= \max_{1 \leq j \leq n} \sum_{1 \leq k \leq r} |b_{kj}| \sum_{1 \leq i \leq m} |a_{ik}| \\
&\leq \|\mathbf{B}\|_1 \max_{1 \leq k \leq r} \sum_{1 \leq i \leq m} |a_{ik}| \\
&= \|\mathbf{B}\|_1 \|\mathbf{A}\|_1 \\
&= \|\mathbf{A}\|_1 \|\mathbf{B}\|_1.
\end{aligned}
$$

Therefore, we have proven that given an arbitrary real matrix $\mathbf{A} = [a_{ij}] \in \mathbb{R}^{m \times n}$, $\|\mathbf{A}\|_1 = \max_{1 \leq j \leq n} \sum_{1 \leq i \leq m} |a_{ij}|$ is a sub-multiplicative norm. $\qquad\square$

**Lemma 4.** *For all $\mathbf{s} \in \mathcal{S}$ and $\mathbf{a} \in \mathcal{A}$, Shapley-Bellman operator is a contraction mapping in a non-empty complete metric space when $\max_{\mathbf{s}} \left\{ \sum_{i \in \mathcal{N}} \max_{a_i} w_i(\mathbf{s}, a_i) \right\} < \frac{1}{\gamma}$.*

*Proof.* The complete proof is as follows.

To ease life, we firstly define some variables that will be used for proof such that

$$
\begin{aligned}
\mathbf{Q}^{\phi} &= \times_{i \in \mathcal{N}} Q_i^{\phi} \in \mathbb{R}^{|\mathcal{N}| \times |\mathcal{S}||\mathcal{A}|}, \\
\mathbf{w} &\in \mathbb{R}^{|\mathcal{N}| \times |\mathcal{S}||\mathcal{A}|}, \\
Pr &\in \mathbb{R}^{|\mathcal{S}||\mathcal{A}| \times |\mathcal{S}|}, \\
\mathbf{1} &= [1, 1, ..., 1]^{\top},
\end{aligned}
$$

where $\mathcal{A} = \times_{i \in \mathcal{N}} \mathcal{A}_i$. Then, for an arbitrary matrix $\mathbf{A} \in \mathbb{R}^{m \times n}$, we define the $\|\cdot\|_1$ for the induced matrix norm such that

$$\|\mathbf{A}\|_1 = \max_{1 \leq j \leq n} \sum_{1 \leq i \leq m} |a_{ij}|,$$

where $a_{ij}$ is an arbitrary element in $\mathbf{A}$. By Lemma 3, $\|\cdot\|_1$ defined here is a sub-multiplicative norm. By Lemma 2, the set of real matrices $\mathbb{R}^{|\mathcal{N}| \times |\mathcal{S}||\mathcal{A}|}$ with the norm $\|\cdot\|_1$ is a Banach algebra and a non-empty complete metric space with the metric induced by $\|\cdot\|_1$.

To show that the operator $\Upsilon$ is a contraction mapping in the supremum norm, we just need to show that for any $\mathbf{Q}_1^{\phi} = \times_{i \in \mathcal{N}} (Q_i^{\phi})_1 \in \mathbb{R}^{|\mathcal{N}| \times |\mathcal{S}||\mathcal{A}|}$ and $\mathbf{Q}_2^{\phi} = \times_{i \in \mathcal{N}} (Q_i^{\phi})_2 \in \mathbb{R}^{|\mathcal{N}| \times |\mathcal{S}||\mathcal{A}|}$, we have

$\|\Upsilon\mathbf{Q}_1^\phi - \Upsilon\mathbf{Q}_2^\phi\|_1 \le \delta\|\mathbf{Q}_1^\phi - \mathbf{Q}_2^\phi\|_1$, where $\delta \in (0,1)$.

$$\|\Upsilon\mathbf{Q}_1^\phi - \Upsilon\mathbf{Q}_2^\phi\|_1$$

$$= \max_{\mathbf{s},\mathbf{a}} \mathbf{1}^\top \Big| \mathbf{w}(\mathbf{s},\mathbf{a}) \sum_{\mathbf{s}'\in\mathcal{S}} Pr(\mathbf{s}'|\mathbf{s},\mathbf{a})\Big[R(\mathbf{s},\mathbf{a}) + \gamma \sum_{i\in\mathcal{N}} \max_{a_i} \big(Q_i^\phi\big)_1(\mathbf{s}',a_i)\Big] - \mathbf{b}(\mathbf{s})$$

$$- \mathbf{w}(\mathbf{s},\mathbf{a}) \sum_{\mathbf{s}'\in\mathcal{S}} Pr(\mathbf{s}'|\mathbf{s},\mathbf{a})\Big[R(\mathbf{s},\mathbf{a}) + \gamma \sum_{i\in\mathcal{N}} \max_{a_i} \big(Q_i^\phi\big)_2(\mathbf{s}',a_i)\Big] + \mathbf{b}(\mathbf{s})\Big|$$

$$= \gamma\max_{\mathbf{s},\mathbf{a}} \mathbf{1}^\top \Big| \mathbf{w}(\mathbf{s},\mathbf{a}) \sum_{\mathbf{s}'\in\mathcal{S}} Pr(\mathbf{s}'|\mathbf{s},\mathbf{a})\Big[ \sum_{i\in\mathcal{N}} \max_{a_i} \big(Q_i^\phi\big)_1(\mathbf{s}',a_i) - \sum_{i\in\mathcal{N}} \max_{a_i} \big(Q_i^\phi\big)_2(\mathbf{s}',a_i)\Big]\Big|$$

$$\le \gamma\max_{\mathbf{s},\mathbf{a}} \mathbf{1}^\top \Big| \mathbf{w}(\mathbf{s},\mathbf{a}) \Big| \max_{\mathbf{s},\mathbf{a}} \Big| \sum_{\mathbf{s}'\in\mathcal{S}} Pr(\mathbf{s}'|\mathbf{s},\mathbf{a})\Big[ \sum_{i\in\mathcal{N}} \max_{a_i} \big(Q_i^\phi\big)_1(\mathbf{s}',a_i) - \sum_{i\in\mathcal{N}} \max_{a_i} \big(Q_i^\phi\big)_2(\mathbf{s}',a_i)\Big]\Big|$$

$$\left(\text{If we write } \delta = \gamma\max_{\mathbf{s},\mathbf{a}} \mathbf{1}^\top\big|\mathbf{w}(\mathbf{s},\mathbf{a})\big|, \text{ we can have the following equation.}\right)$$

$$= \delta\max_{\mathbf{s},\mathbf{a}} \Big| \sum_{\mathbf{s}'\in\mathcal{S}} Pr(\mathbf{s}'|\mathbf{s},\mathbf{a})\Big[ \sum_{i\in\mathcal{N}} \max_{a_i} \big(Q_i^\phi\big)_1(\mathbf{s}',a_i) - \sum_{i\in\mathcal{N}} \max_{a_i} \big(Q_i^\phi\big)_2(\mathbf{s}',a_i)\Big]\Big|$$

$$\le \delta\max_{\mathbf{s},\mathbf{a}} \sum_{\mathbf{s}'\in\mathcal{S}} Pr(\mathbf{s}'|\mathbf{s},\mathbf{a})\Big| \sum_{i\in\mathcal{N}} \max_{a_i} \big(Q_i^\phi\big)_1(\mathbf{s}',a_i) - \sum_{i\in\mathcal{N}} \max_{a_i} \big(Q_i^\phi\big)_2(\mathbf{s}',a_i)\Big|$$

$$= \delta\Big| \sum_{i\in\mathcal{N}}\Big[ \max_{a_i} \big(Q_i^\phi\big)_1(\mathbf{s}',a_i) - \max_{a_i} \big(Q_i^\phi\big)_2(\mathbf{s}',a_i)\Big]\Big|$$

(By triangle inequality, we can obtain the following inequality.)

$$\le \delta \sum_{i\in\mathcal{N}} \Big| \max_{a_i} \big(Q_i^\phi\big)_1(\mathbf{s}',a_i) - \max_{a_i} \big(Q_i^\phi\big)_2(\mathbf{s}',a_i)\Big|$$

$$\le \delta \sum_{i\in\mathcal{N}} \max_{a_i} \Big| \big(Q_i^\phi\big)_1(\mathbf{s}',a_i) - \big(Q_i^\phi\big)_2(\mathbf{s}',a_i)\Big|$$

(Since $\mathbf{a} = \times_{i\in\mathcal{N}} a_i$, we have the following equation.)

$$= \delta \max_{\mathbf{a}} \sum_{i\in\mathcal{N}} \Big| \big(Q_i^\phi\big)_1(\mathbf{s}',a_i) - \big(Q_i^\phi\big)_2(\mathbf{s}',a_i)\Big|$$

$$\le \delta \max_{\mathbf{z},\mathbf{a}} \sum_{i\in\mathcal{N}} \Big| \big(Q_i^\phi\big)_1(\mathbf{z},a_i) - \big(Q_i^\phi\big)_2(\mathbf{z},a_i)\Big| = \delta\|\mathbf{Q}_1^\phi - \mathbf{Q}_2^\phi\|_1.$$

Now, we need to discuss the condition to $\delta \in (0,1)$. Apparently, $\delta > 0$, so we just need to discuss the condition to guarantee that $\delta < 1$. We now have the following discussions such that

$$\delta = \gamma\max_{\mathbf{s},\mathbf{a}} \mathbf{1}^\top\big|\mathbf{w}(\mathbf{s},\mathbf{a})\big| < 1 \text{ (Since } w_i(\mathbf{s},a_i) > 0.)$$

$$\Rightarrow \gamma\max_{\mathbf{s},\mathbf{a}} \sum_{i\in\mathcal{N}} w_i(\mathbf{s},a_i) < 1$$

(When $\gamma \ne 0$, we can have the following inequality.)

$$\Rightarrow \max_{\mathbf{s},\mathbf{a}} \sum_{i\in\mathcal{N}} w_i(\mathbf{s},a_i) < \frac{1}{\gamma}$$

(Since $\mathbf{a} = \times_{i\in\mathcal{N}} a_i$, we have the following equation.)

$$\Rightarrow \max_{\mathbf{s}} \Big\{ \sum_{i\in\mathcal{N}} \max_{a_i} w_i(\mathbf{s},a_i)\Big\} < \frac{1}{\gamma}.$$

Therefore, we show that Shapley-Bellman operator $\Upsilon$ is a contraction mapping in the non-empty complete metric space generated by $\mathbb{R}^{|\mathcal{N}|\times|\mathcal{S}||\mathcal{A}|}$ with the metric induced by $\|\cdot\|_1$, when $\max_{\mathbf{s}}\Big\{ \sum_{i\in\mathcal{N}} \max_{a_i} w_i(\mathbf{s},a_i)\Big\} < \frac{1}{\gamma}$. Finally, it is apparent that $w_i(\mathbf{s},a_i) = 1/|\mathcal{N}|$ when $a_i = \arg\max_{a_i} Q_i^\phi(\mathbf{s},a_i)$ satisfies the above condition. $\qquad\square$

**Corollary 1.** *According to Banach fixed-point theorem [46], Shapley-Bellman operator admits a unique fixed point. Moreover, starting by an arbitrary start point, the sequence recursively generated by Shapley-Bellman operator can finally converge to that fixed point.*

*Proof.* Since $\langle \mathbb{R}^{|\mathcal{N}| \times |\mathcal{S}||\mathcal{A}|}, ||\cdot||_1 \rangle$ is a non-empty complete metric space and Shapley-Bellman operator $\Upsilon$ is shown as a contraction mapping in Lemma 4, by Banach fixed-point theorem [46] we can directly conclude that Shapley-Bellman operator $\Upsilon$ admits a unique fixed point. Furthermore, starting by an arbitrary start point, the sequence recursively generated by Shapley-Bellman operator $\Upsilon$ can finally converge to that fixed point. $\qquad\square$

**Theorem 1.** *Shapley-Bellman operator can converge to the optimal Markov Shapley Q-value and the corresponding optimal joint deterministic policy when $\max_{\mathbf{s}} \left\{ \sum_{i \in \mathcal{N}} \max_{a_i} w_i(\mathbf{s}, a_i) \right\} < \frac{1}{\gamma}$.*

*Proof.* By Corollary 1, we get that Shapley-Bellman operator admits a unique fixed point. Since Shapley-Bellman optimality equation (i.e., Eq.7) is obviously a fixed point for Shapley-Bellman operator, it is not difficult to get the conclusion that the optimal Markov Shapley Q-value is achieved. Since the sum of optimal Markov Shapley Q-values is equal to the optimal global Q-value and the optimal global Q-value corresponds to the optimal joint deterministic policy, we show that the optimal joint deterministic policy is achieved. Besides, it is obvious that Shapley-Bellman optimality equation can be transformed back to the Bellman optimality equation w.r.t. the optimal global Q-value, given the efficiency property of Markov Shapley value. $\qquad\square$

### E.4.3 Stochastic Approximation of Shapley-Bellman operator

We now derive the stochastic approximation of Shapley-Bellman operator over the value space, i.e. a form of Q-learning derived from Shapley-Bellman operator. By sampling from $Pr(\mathbf{s}'|\mathbf{s}, \mathbf{a})$ via Monte Carlo method, the Q-learning algorithm can be expressed as follows:

$$\mathbf{Q}_{t+1}^{\phi}(\mathbf{s}, \mathbf{a}) \leftarrow \mathbf{Q}_t^{\phi}(\mathbf{s}, \mathbf{a}) + \alpha_t(\mathbf{s}, \mathbf{a}) \Big[ \mathbf{w}(\mathbf{s}, \mathbf{a}) \big( R_t + \gamma \sum_{i \in \mathcal{N}} \max_{a_i} (Q_i^{\phi})_t(\mathbf{s}', a_i) \big) - \mathbf{b}(\mathbf{s}) - \mathbf{Q}_t^{\phi}(\mathbf{s}, \mathbf{a}) \Big]. \quad (55)$$

**Lemma 5** (Jaakkola et al. [47]). *The random process $\{\Delta_t\}$ taking values $\mathbb{R}^n$ defined as*

$$\Delta_{t+1}(x) = (1 - \alpha_t(x))\Delta_t(x) + \alpha_t(x)F_t(x)$$

*converges to 0 w.p.1 under the following assumptions:*

- $0 \le \alpha_t \le 1$, $\sum_t \alpha_t(x) = \infty$ and $\sum_t \alpha_t^2 \le \infty$;
- $\|\mathbb{E}[F_t(x)|\mathcal{F}_t]\|_W \le \delta \|\Delta_t\|_W$, *with* $0 \le \delta < 1$;
- $var[F_t(x)|\mathcal{F}_t] \le C(1 + \|\Delta_t\|_W^2)$, *for* $C > 0$.

**Theorem 4.** *For a finite Markov convex game, the Q-learning algorithm derived by Shapley-Bellman operator given by the update rule such that*

$$\mathbf{Q}_{t+1}^{\phi}(\mathbf{s}, \mathbf{a}) \leftarrow \mathbf{Q}_t^{\phi}(\mathbf{s}, \mathbf{a}) + \alpha_t(\mathbf{s}, \mathbf{a}) \left[ \mathbf{w}(\mathbf{s}, \mathbf{a}) \left( R_t + \gamma \sum_{i \in \mathcal{N}} \max_{a_i} (Q_i^{\phi})_t(\mathbf{s}', a_i) \right) - \mathbf{b}(\mathbf{s}) - \mathbf{Q}_t^{\phi}(\mathbf{s}, \mathbf{a}) \right],$$

*converges w.p.1 to the optimal Markov Shapley Q-value if*

$$\sum_t \alpha_t(\mathbf{s}, \mathbf{a}) = \infty \qquad \sum_t \alpha_t^2(\mathbf{s}, \mathbf{a}) \le \infty \qquad (56)$$

*for all $\mathbf{s} \in \mathcal{S}$ and $\mathbf{a} \in \mathcal{A}$ as well as $\max_{\mathbf{s}} \left\{ \sum_{i \in \mathcal{N}} \max_{a_i} w_i(\mathbf{s}, a_i) \right\} < \frac{1}{\gamma}$.*

*Proof.* The proof follows the sketch of proving the convergence of Q-learning given by Melo [48]. First, we rewrite Eq.55 to

$$\mathbf{Q}_t^{\phi}(\mathbf{s}, \mathbf{a}) = (1 - \alpha_t(\mathbf{s}, \mathbf{a}))\,\mathbf{Q}_t^{\phi}(\mathbf{s}, \mathbf{a}) + \alpha_t(\mathbf{s}, \mathbf{a}) \left[ \mathbf{w}(\mathbf{s}, \mathbf{a}) \left( R_t + \gamma \sum_{i \in \mathcal{N}} \max_{a_i} (Q_i^{\phi})_t(\mathbf{s}', a_i) \right) - \mathbf{b}(\mathbf{s}) \right].$$

By subtracting $\mathbf{Q}^{\phi^*}(\mathbf{s}, \mathbf{a})$ and letting

$$\Delta_t(\mathbf{s}, \mathbf{a}) = \mathbf{Q}_t^{\phi}(\mathbf{s}, \mathbf{a}) - \mathbf{Q}^{\phi^*}(\mathbf{s}, \mathbf{a}),$$

we can transform Eq.55 to

$$\Delta_{t+1}(\mathbf{s},\mathbf{a}) = (1 - \alpha_t(\mathbf{s},\mathbf{a}))\Delta_t(\mathbf{s},\mathbf{a}) + \alpha_t(\mathbf{s},\mathbf{a})F_t(\mathbf{s},\mathbf{a}),$$

where

$$F_t(\mathbf{s},\mathbf{a}) = \mathbf{w}(\mathbf{s},\mathbf{a})\left(R_t + \gamma \sum_{i\in\mathcal{N}} \max_{a_i}(Q_i^\phi)_t(\mathbf{s}',a_i)\right) - \mathbf{b}(\mathbf{s}) - \mathbf{Q}^{\phi^*}(\mathbf{s},\mathbf{a}).$$

Since $\mathbf{s}' \in \mathcal{S}$ is a random sample from Markov Chain, so we can get that

$$\mathbb{E}[F_t(\mathbf{s},\mathbf{a})|\mathcal{F}_t] = \sum_{\mathbf{s}'\in\mathcal{S}} Pr(\mathbf{s}'|\mathbf{s},\mathbf{a})\left[\mathbf{w}(\mathbf{s},\mathbf{a})\left(R_t + \gamma \sum_{i\in\mathcal{N}} \max_{a_i}(Q_i^\phi)_t(\mathbf{s}',a_i)\right) - \mathbf{b}(\mathbf{s}) - \mathbf{Q}^{\phi^*}(\mathbf{s},\mathbf{a})\right]$$

$$= \mathbf{w}(\mathbf{s},\mathbf{a})\sum_{\mathbf{s}'\in\mathcal{S}} Pr(\mathbf{s}'|\mathbf{s},\mathbf{a})\left(R_t + \gamma \sum_{i\in\mathcal{N}} \max_{a_i}(Q_i^\phi)_t(\mathbf{s}',a_i)\right) - \mathbf{b}(\mathbf{s}) - \mathbf{Q}^{\phi^*}(\mathbf{s},\mathbf{a})$$

$$\left(\text{Since } \max_{\mathbf{s}}\left\{\sum_{i\in\mathcal{N}} \max_{a_i} w_i(\mathbf{s},a_i)\right\} < \frac{1}{\gamma}.\right)$$

$$= \Upsilon\mathbf{Q}_t^\phi(\mathbf{s},\mathbf{a}) - \Upsilon\mathbf{Q}^{\phi^*}(\mathbf{s},\mathbf{a}).$$

By the results from Theorem 4, we can get that

$$\|\mathbb{E}[F_t(\mathbf{s},\mathbf{a})|\mathcal{F}_t]\|_1 \le \delta\|\mathbf{Q}_t^\phi(\mathbf{s},\mathbf{a}) - \mathbf{Q}^{\phi^*}(\mathbf{s},\mathbf{a})\|_1 = \delta\|\Delta_t(\mathbf{s},\mathbf{a})\|_1,$$

where $\delta \in (0,1)$.

Next, we get that

$$\mathbf{var}[F_t(\mathbf{s},\mathbf{a})|\mathcal{F}_t] = \mathbb{E}\left[\left(\mathbf{w}(\mathbf{s},\mathbf{a})\left(R_t + \gamma \sum_{i\in\mathcal{N}} \max_{a_i}(Q_i^\phi)_t(\mathbf{s}',a_i)\right) - \mathbf{b}(\mathbf{s}) - \mathbf{Q}^{\phi^*}(\mathbf{s},\mathbf{a})\right.\right.$$

$$\left.\left. - \Upsilon\mathbf{Q}_t^\phi(\mathbf{s},\mathbf{a}) + \mathbf{Q}^{\phi^*}(\mathbf{s},\mathbf{a})\right)^2\right]$$

$$= \mathbb{E}\left[\left(\mathbf{w}(\mathbf{s},\mathbf{a})\left(R_t + \gamma \sum_{i\in\mathcal{N}} \max_{a_i}(Q_i^\phi)_t(\mathbf{s}',a_i)\right) - \mathbf{b}(\mathbf{s}) - \Upsilon\mathbf{Q}_t^\phi(\mathbf{s},\mathbf{a})\right)^2\right]$$

$$= \mathbf{var}\left[\mathbf{w}(\mathbf{s},\mathbf{a})\left(R_t + \gamma \sum_{i\in\mathcal{N}} \max_{a_i}(Q_i^\phi)_t(\mathbf{s}',a_i)\right) - \mathbf{b}(\mathbf{s})|\mathcal{F}_t\right].$$

Since $R_t$, $\mathbf{w}(\mathbf{s},\mathbf{a})$ and $\mathbf{b}(\mathbf{s})$ are bounded, it clearly verifies that

$$\mathbf{var}[F_t(\mathbf{s},\mathbf{a})|\mathcal{F}_t] \le C(1 + \|\Delta_t(\mathbf{s},\mathbf{a})\|_1^2)$$

for some constant $C$.

Finally, by Lemma 5 it is easy to see that $\Delta_t$ converges to 0 w.p.1, i.e., $\mathbf{Q}_t^\phi(\mathbf{s},\mathbf{a})$ converges to $\mathbf{Q}^{\phi^*}(\mathbf{s},\mathbf{a})$ w.p.1, given the condition in Eq.56. $\square$

### E.4.4 Derivation of Shapley Q-Learning

Similar to the operations in Section E.4.3, by stochastic approximation in value space, i.e. sampling $\mathbf{s}'$ from $Pr(\mathbf{s}'|\mathbf{s},\mathbf{a})$ via Monte Carlo method, Shapley-Bellman operator can be expressed as follows:

$$\mathbf{Q}^\phi(\mathbf{s},\mathbf{a}) = \mathbf{w}(\mathbf{s},\mathbf{a})\left(R + \gamma \sum_{i\in\mathcal{N}} \max_{a_i} Q_i^\phi(\mathbf{s}',a_i)\right) - \mathbf{b}(\mathbf{s}), \tag{57}$$

where $\mathbf{w}(\mathbf{s},\mathbf{a}) = [w_i(\mathbf{s},a_i)]^\top \in \mathbb{R}_+^{|\mathcal{N}|}$; $\mathbf{b}(\mathbf{s}) = [b_i(\mathbf{s})]^\top \in \mathbb{R}_+^{|\mathcal{N}|}$; and $\mathbf{Q}^\phi(\mathbf{s},\mathbf{a}) = [Q_i^\phi(\mathbf{s},a_i)]^\top \in \mathbb{R}_+^{|\mathcal{N}|}$. Since $\mathbf{w}(\mathbf{s},\mathbf{a}) = diag(\mathbf{w}(\mathbf{s},\mathbf{a}))\,\mathbf{1}$ where $diag(\cdot)$ denotes the diagonalization of a vector[8] and $\mathbf{1}$ denotes the vector of ones, Eq.57 can be equivalently represented as

$$\mathbf{Q}^\phi(\mathbf{s},\mathbf{a}) = diag(\mathbf{w}(\mathbf{s},\mathbf{a}))\,\mathbf{1}\left(R + \gamma \sum_{i\in\mathcal{N}} \max_{a_i} Q_i^\phi(\mathbf{s}',a_i)\right) - \mathbf{b}(\mathbf{s}). \tag{58}$$

---

[8]It is a square diagonal matrix with the elements of vector v on the main diagonal, and the other entries of the matrix are zeros.

Since $w_i(\mathbf{s}, a_i) > 0, \forall i \in \mathcal{N}$, we can write the following equivalent form to Eq.58 such that

$$diag\big(\mathbf{w}(\mathbf{s}, \mathbf{a})\big)^{-1}\mathbf{Q}^\phi(\mathbf{s}, \mathbf{a}) = \mathbf{1} \left( R + \gamma \sum_{i \in \mathcal{N}} \max_{a_i} Q_i^\phi(\mathbf{s}', a_i) \right) - diag\big(\mathbf{w}(\mathbf{s}, \mathbf{a})\big)^{-1}\mathbf{b}(\mathbf{s}). \tag{59}$$

Next, we multiply $\mathbf{1}^\top$ on both sides and obtain the following equation such that

$$\sum_{i \in \mathcal{N}} \frac{1}{w_i(\mathbf{s}, a_i)} Q_i^\phi(\mathbf{s}, a_i) = |\mathcal{N}| \left( R + \gamma \sum_{i \in \mathcal{N}} \max_{a_i} Q_i^\phi(\mathbf{s}', a_i) \right) - \sum_{i \in \mathcal{N}} w_i(\mathbf{s}, a_i)^{-1} b_i(\mathbf{s}). \tag{60}$$

Since the condition such that $\sum_{i \in \mathcal{N}} w_i(\mathbf{s}, a_i)^{-1} b_i(\mathbf{s}) = 0$, by dividing $|\mathcal{N}|$ on both sides we get that

$$\sum_{i \in \mathcal{N}} \frac{1}{|\mathcal{N}| w_i(\mathbf{s}, a_i)} Q_i^\phi(s, a_i) = R + \gamma \sum_{i \in \mathcal{N}} \max_{a_i} Q_i^\phi(s, a_i). \tag{61}$$

Since $w_i(\mathbf{s}, a_i) = 1/|\mathcal{N}|$ when $a_i = \arg\max_{a_i} Q_i^\phi(\mathbf{s}, a_i)$, by defining $\delta_i(\mathbf{s}, a_i) = \frac{1}{|\mathcal{N}| w_i(\mathbf{s}, a_i)}$ we can get that

$$\delta_i(\mathbf{s}, a_i) = \begin{cases} 1 & a_i = \arg\max_{a_i} Q_i^\phi(\mathbf{s}, a_i), \\ \alpha_i(\mathbf{s}, a_i) & a_i \neq \arg\max_{a_i} Q_i^\phi(\mathbf{s}, a_i), \end{cases} \tag{62}$$

where $\alpha_i(\mathbf{s}, a_i)$ is a variable that expresses $\frac{1}{|\mathcal{N}| w_i(\mathbf{s}, a_i)}$ when $a_i \neq \arg\max_{a_i} Q_i^\phi(\mathbf{s}, a_i)$ for the ease of implementation.

Substituting Eq.62 into Eq.61, we can get the following equation such that

$$\sum_{i \in \mathcal{N}} \delta_i(\mathbf{s}, a_i) Q_i^\phi(\mathbf{s}, a_i) = R + \gamma \sum_{i \in \mathcal{N}} \max_{a_i} Q_i^\phi(\mathbf{s}', a_i). \tag{63}$$

By rearranging Eq.63, we obtain the TD error of Shapley Q-learning (SHAQ) such that

$$\Delta(\mathbf{s}, \mathbf{a}, \mathbf{s}') = R + \gamma \sum_{i \in \mathcal{N}} \max_{a_i} Q_i^\phi(\mathbf{s}', a_i) - \sum_{i \in \mathcal{N}} \delta_i(\mathbf{s}, a_i) Q_i^\phi(\mathbf{s}, a_i). \tag{64}$$

Note that the TD error of SHAQ is necessary for the TD error of Eq.55 (i.e. the stochastic learning process that we proved to converge to the optimal Markov Shapley Q-value in Theorem 4). For this reason, the condition $\max_{\mathbf{s}} \left\{ \sum_{i \in \mathcal{N}} \max_{a_i} w_i(\mathbf{s}, a_i) \right\} < \frac{1}{\gamma}$ is necessary to be satisfied so that the convergence to the optimality is possible to hold.

### E.5 Mathematical Proofs of Validity and Interpretability

**Lemma 6.** *Markov core is a convex set.*

*Proof.* Let $\big( \max_{\pi_i} x_i(\mathbf{s}) \big)_{i \in \mathcal{N}}$ and $\big( \max_{\pi_i} y_i(\mathbf{s}) \big)_{i \in \mathcal{N}}$ be two vectors in the Markov core and $\alpha \in [0, 1)$ be an arbitrary scalar. To ease life, for any $i \in \mathcal{N}$ we let $\max_{\pi_i} z_i(\mathbf{s}) = \alpha \max_{\pi_i} x_i(\mathbf{s}) + (1 - \alpha) \max_{\pi_i} y_i(\mathbf{s})$. By definition, for any coalition $\mathcal{C} \subseteq \mathcal{N}$ we have

$$\begin{aligned} \max_{\pi_\mathcal{C}} z(\mathbf{s}|\mathcal{C}) &= \sum_{i \in \mathcal{C}} \max_{\pi_i} z_i(\mathbf{s}) \\ &= \sum_{i \in \mathcal{C}} \alpha \max_{\pi_i} x_i(\mathbf{s}) + (1 - \alpha) \max_{\pi_i} y_i(\mathbf{s}) \\ &= \alpha \sum_{i \in \mathcal{C}} \max_{\pi_i} x_i(\mathbf{s}) + (1 - \alpha) \sum_{i \in \mathcal{C}} \max_{\pi_i} y_i(\mathbf{s}) \\ &\geq \alpha \max_{\pi_\mathcal{C}} V^{\pi_\mathcal{C}}(\mathbf{s}) + (1 - \alpha) \max_{\pi_\mathcal{C}} V^{\pi_\mathcal{C}}(\mathbf{s}) \\ &= \max_{\pi_\mathcal{C}} V^{\pi_\mathcal{C}}(\mathbf{s}). \end{aligned}$$

Therefore, we proved that Markov core is a convex set. $\qquad\square$

**Theorem 2.** *The optimal Markov Shapley value is a solution in the Markov core under Markov convex game (MCG) with the grand coalition.*

*Proof.* The optimal Markov Shapley value is the affine combination of the optimal marginal contributions. We know that Markov core is a convex set by Lemma 6 and the optimal marginal contribution is in the Markov core by Lemma 1. Since the affine combination of the points in a convex set is still in this convex set, we get that the optimal Markov Shapley value is in the Markov core. $\qquad\square$

### E.6 Mathematical Derivation for Implementation of Shapley Q-Learning

**Proposition 3.** *Suppose any action marginal contribution can be factorised to the form such that* $\Phi_i(\mathbf{s}, a_i | \mathcal{C}_i) = \sigma(\mathbf{s}, \mathbf{a}_{\mathcal{C}_i \cup \{i\}}) \, \hat{Q}_i(\mathbf{s}, a_i)$. *With the condition such that*

$$\mathbb{E}_{\mathcal{C}_i \sim Pr(\mathcal{C}_i | \mathcal{N} \setminus \{i\})} [\sigma(\mathbf{s}, \mathbf{a}_{\mathcal{C}_i \cup \{i\}})] = \begin{cases} 1 & a_i = \arg\max_{a_i} Q_i^\phi(\mathbf{s}, a_i), \\ K \in (0, 1) & a_i \neq \arg\max_{a_i} Q_i^\phi(\mathbf{s}, a_i), \end{cases}$$

*we have*

$$\begin{cases} Q_i^\phi(\mathbf{s}, a_i) = \hat{Q}_i(\mathbf{s}, a_i) & a_i = \arg\max_{a_i} \hat{Q}_i(\mathbf{s}, a_i), \\ \alpha_i(\mathbf{s}, a_i) \, Q_i^\phi(\mathbf{s}, a_i) = \hat{\alpha}_i(\mathbf{s}, a_i) \, \hat{Q}_i(\mathbf{s}, a_i) & a_i \neq \arg\max_{a_i} \hat{Q}_i(\mathbf{s}, a_i), \end{cases}$$

*where* $\hat{\alpha}_i(\mathbf{s}, a_i) = \mathbb{E}_{\mathcal{C}_i \sim Pr(\mathcal{C}_i | \mathcal{N} \setminus \{i\})} [\hat{\psi}_i(\mathbf{s}, a_i; \mathbf{a}_{\mathcal{C}_i})]$ *and* $\hat{\psi}_i(\mathbf{s}, a_i; \mathbf{a}_{\mathcal{C}_i}) := \alpha_i(\mathbf{s}, a_i) \, \sigma(\mathbf{s}, \mathbf{a}_{\mathcal{C}_i \cup \{i\}})$.

*Proof.* We suppose for any $\mathbf{s} \in \mathcal{S}$ and $\mathbf{a} \in \mathcal{A}$, we have $\Phi_i(\mathbf{s}, a_i | \mathcal{C}_i) = \sigma(\mathbf{s}, \mathbf{a}_{\mathcal{C}_i \cup \{i\}}) \, \hat{Q}_i(\mathbf{s}, a_i)$ and $\mathbb{E}_{\mathcal{C}_i} [\sigma(\mathbf{s}, \mathbf{a}_{\mathcal{C}_i \cup \{i\}})] = 1$ when $a_i = \arg\max_{a_i} Q_i^\phi(\mathbf{s}, a_i)$. By the definition of the Markov Shapley Q-value, it is not difficult to obtain

$$\begin{aligned} Q_i^\phi(\mathbf{s}, a_i) &= \mathbb{E}_{\mathcal{C}_i} [\Phi_i(\mathbf{s}, a_i | \mathcal{C}_i)] \\ &= \mathbb{E}_{\mathcal{C}_i} [\sigma(\mathbf{s}, \mathbf{a}_{\mathcal{C}_i \cup \{i\}}) \, \hat{Q}_i(\mathbf{s}, a_i)] \\ &= \mathbb{E}_{\mathcal{C}_i} [\sigma(\mathbf{s}, \mathbf{a}_{\mathcal{C}_i \cup \{i\}})] \, \hat{Q}_i(\mathbf{s}, a_i). \end{aligned}$$

Recall that $\delta_i(\mathbf{s}, a_i)$ is defined as follows:

$$\delta_i(\mathbf{s}, a_i) = \begin{cases} 1 & a_i = \arg\max_{a_i} Q_i^\phi(\mathbf{s}, a_i), \\ \alpha_i(\mathbf{s}, a_i) & a_i \neq \arg\max_{a_i} Q_i^\phi(\mathbf{s}, a_i). \end{cases}$$

If $a_i = \arg\max_{a_i} Q_i^\phi(\mathbf{s}, a_i)$, it is not difficult to get that $Q_i^\phi(\mathbf{s}, a_i) = \hat{Q}_i(\mathbf{s}, a_i)$.

If $a_i \neq \arg\max_{a_i} Q_i^\phi(\mathbf{s}, a_i)$, we can have the following equation such that

$$\begin{aligned} \alpha_i(\mathbf{s}, a_i) \, Q_i^\phi(\mathbf{s}, a_i) &= \alpha_i(\mathbf{s}, a_i) \, \mathbb{E}_{\mathcal{C}_i} [\sigma(\mathbf{s}, \mathbf{a}_{\mathcal{C}_i \cup \{i\}}) \, \hat{Q}_i(\mathbf{s}, a_i)] \\ &= \mathbb{E}_{\mathcal{C}_i} [\alpha_i(\mathbf{s}, a_i) \, \sigma(\mathbf{s}, \mathbf{a}_{\mathcal{C}_i \cup \{i\}})] \, \hat{Q}_i(\mathbf{s}, a_i) \\ &:= \mathbb{E}_{\mathcal{C}_i} [\hat{\psi}_i(\mathbf{s}, a_i; \mathbf{a}_{\mathcal{C}_i})] \, \hat{Q}_i(\mathbf{s}, a_i), \end{aligned}$$

where $\alpha_i(\mathbf{s}, a_i) \, \sigma(\mathbf{s}, \mathbf{a}_{\mathcal{C}_i \cup \{i\}})$ is defined as $\hat{\psi}_i(\mathbf{s}, a_i; \mathbf{a}_{\mathcal{C}_i})$. Since under this situation $\hat{Q}_i(\mathbf{s}, a_i)$ is always a scaled $Q_i^\phi(\mathbf{s}, a_i)$ with the scale of $1/K$, the decisions are consistent to the original decisions. $\square$

### E.6.1 Implementation of $\hat{\alpha}_i(\mathbf{s}, a_i)$

As introduced in the main part of paper, when $a_i \neq \arg\max_{a_i} \hat{Q}_i(\mathbf{s}, a_i)$, $\hat{\alpha}_i(\mathbf{s}, a_i)$ is implemented as follows:

$$\hat{\alpha}_i(\mathbf{s}, a_i) = \frac{1}{M} \sum_{k=1}^M F_{\mathbf{s}} \Big( \hat{Q}_{\mathcal{C}_i^k}(\tau_{\mathcal{C}_i^k}, \mathbf{a}_{\mathcal{C}_i^k}), \, \hat{Q}_i(\tau_i, a_i) \Big) + 1,$$

where

$$\hat{Q}_{\mathcal{C}_i^k}(\tau_{\mathcal{C}_i^k}, \mathbf{a}_{\mathcal{C}_i^k}) = \frac{1}{|\mathcal{C}_i^k|} \sum_{j \in \mathcal{C}_i^k} \hat{Q}_j(\tau_j, a_j)$$

and $\mathcal{C}_i^k \sim Pr(\mathcal{C}_i | \mathcal{N} \setminus \{i\})$ that follows the distribution w.r.t. the occurrence frequency of $\mathcal{C}_i$; and $F_{\mathbf{s}}(\cdot, \cdot)$ is a monotonic function with an absolute activation function on the output whose weights are generated from hypernetworks w.r.t. the global state, similar to the architecture of QMIX [9]. Since $F_{\mathbf{s}}(\cdot, \cdot) \geq 0$ always holds, it is not difficult to obtain that $\hat{\alpha}_i(\mathbf{s}, a_i) \geq 1$ always holds. As Eq.11 shows, it is not difficult to get that $\alpha_i(\mathbf{s}, a_i) = K^{-1} \hat{\alpha}_i(\mathbf{s}, a_i)$. Since $K \in (0, 1)$, we get that $\alpha_i(\mathbf{s}, a_i) > 1$.

As introduced in the main part of paper, the following equation is satisfied such that

$$\delta_i(\mathbf{s}, a_i) = \frac{1}{|\mathcal{N}| \, w_i(\mathbf{s}, a_i)}.$$

For all $\mathbf{s} \in \mathcal{S}$ and $a_i \neq \arg\max_{a_i} \hat{Q}_i(\mathbf{s}, a_i)$, $\delta_i(\mathbf{s}, a_i) = \alpha_i(\mathbf{s}, a_i) > 1$. So, we can derive that

$$w_i(\mathbf{s}, a_i) = \frac{1}{|\mathcal{N}|\, \alpha_i(\mathbf{s}, a_i)}$$

$$\Rightarrow \max_{a_i} w_i(\mathbf{s}, a_i) = \max_{a_i} \frac{1}{|\mathcal{N}|\, \alpha_i(\mathbf{s}, a_i)} = \frac{1}{|\mathcal{N}|\, \min_{a_i} \alpha_i(\mathbf{s}, a_i)} < \frac{1}{|\mathcal{N}|}$$

$$\Rightarrow 0 < \sum_{i \in \mathcal{N}} \max_{a_i} w_i(\mathbf{s}, a_i) < 1.$$

For all $\mathbf{s} \in \mathcal{S}$ and $a_i = \arg\max_{a_i} \hat{Q}_i(\mathbf{s}, a_i)$, $\delta_i(\mathbf{s}, a_i) = \hat{\delta}_i(\mathbf{s}, a_i) = 1$. So, we can derive that

$$w_i(\mathbf{s}, a_i) = \frac{1}{|\mathcal{N}|}$$

$$\Rightarrow \sum_{i \in \mathcal{N}} \max_{a_i} w_i(\mathbf{s}, a_i) = 1.$$

Therefore, we can directly obtain that for all $\mathbf{s} \in \mathcal{S}$ and $\mathbf{a} \in \mathcal{A}$,

$$0 < \max_{\mathbf{s}} \left\{ \sum_{i \in \mathcal{N}} \max_{a_i} w_i(\mathbf{s}, a_i) \right\} \leq 1.$$

Since $\gamma \in (0, 1)$, we can get that $\frac{1}{\gamma} > 1$. As a result, we show that for all $\mathbf{s} \in \mathcal{S}$ and $\mathbf{a} \in \mathcal{A}$,

$$0 < \max_{\mathbf{s}} \left\{ \sum_{i \in \mathcal{N}} \max_{a_i} w_i(\mathbf{s}, a_i) \right\} < \frac{1}{\gamma}.$$

We get that our implementation of $\hat{\alpha}_i(\mathbf{s}, a_i)$ satisfies the condition in Theorem 1.

## F Potential Negative Societal Impacts

Although this paper studies a fundamental theory of multi-agent reinforcement learning, if the proposed algorithm is applied to real-world applications in the future, there may still exist some potential negative societal impacts. First, since the theory does not consider robustness, it is possible that the proposed algorithm would be attacked or vulnerable to some extreme scenarios like most of machine learning models and algorithms. Fortunately, our theory is orthogonal to the robustness and it is possible to consider robustness as an extension in the future work. Another potential negative societal impacts could come from the implementation of models, e.g., policy and critic. Since these are implemented by neural networks that are known as black boxes, the reliability could be a problem. Nevertheless, this is irrelevant to the main purpose of this paper and can be improved by other related research tracks in the future.