# OpenReview forum: "SHAQ: Incorporating Shapley Value Theory into Multi-Agent Q-Learning"
_NeurIPS.cc/2022/Conference — NeurIPS 2022 Accept_

### Official Review · Reviewer_XnpC · 2022-07-10

**Rating:** 6
**Confidence:** 4
**Soundness:** 3 good
**Presentation:** 2 fair
**Contribution:** 3 good

**Summary:**

The paper considers multiagent reinforcement learning in a global (cooeprative) reward game. It contrasts the results value factorization frameworks, and proposes an alternative via the Shapley value from cooperative game theory. Basically, the authors consider a form of game with coalition structures, and apply the Shapley value to decompose the reward, and derive and Shapley-Bellman optimiality equation (SBOE) corresponding to the optimal joint determinisitic policy. They propose a Shapley-Bellman opeator (SBO) that solves for the SBOE. These finally give rise to a new multiagent reinforcement learning algorithm, called Shapley Q learning, SHAQ for short,  somewhat akin to existing value factorization methods.

Empirically, on a few settings (predator-prey and starcraft)  SHAQ exhibits better performance than existing approaches, and also provides some interpertability foundation.

**Questions:**

What happens in domains which are not completely cooperative (team reward), such as social dillemas or mixed motive games. Does the algorithm still runs? Does it fail? What happens when agents are trained using a mixture of algorithms (SHAQ agent with other approaches - are they compatible). Can you replace the Shapley value with other cooperative solution concepts (the Banzhaf index seems to be the closest, basically your equation just with different weights for the subteams) - does the whole method fail?

**Limitations:**

As I wrote, the empirical analysis is somewhat limited (but certainly a decent foundation). Also the writing could be improved - at the very least I'd give the formal definitions of a transferable utility cooperative game, coalition structures, the core (as applied to a general CS or characteristic function game). The paper does a better job on the RL side (where things are fully defined). Also, you should have the discussion on what happens in non team reward (non fully cooperative) settings.

All in all, a very interesting paper, if only for the nice connection between RL and cooperative game theory.

**Strengths And Weaknesses:**

The key strength of the paper is in applying cooperative game theory tools to multiagent Q learning. Recently the Shapley value has become a very popular tool in machine learning due to its ability to decompose the performance of a model to the relative influence of specific features. This has proven a very strong tool for analyzing supervised learning models. The authors propose now propose to use this theoretic foundation to multiagent reinforcement learning.

The key weakness in my opion is not having a clear, crisp takeaway from this work. If the main claim is the superior performanc on multiagent reinforcement learning, then the empirical analysis seems to be somewhat lacking, as it covers relatively few domains (there are now enough multiagent gyms that allow a wider variability of tasks). If the main claim is a theoretical foundation, then one might expect a tighter analysis and bounds as compared to existing approaches.

Either way, I think the writing is very formal, and could be imporved. What is the main driving intuition here? MARL is typically considered through a non-cooperative game theory prism (Markov Game). Here you are trying to use cooperative game theory, which means you consider subsets of agents, and have some function mapping each such subteam to its success in the task. Then, one might view the Shapley value as a decomposition allocating each single agent its individual reward / impact in the team's success. But why use the Shapley value rather than other solution concepts (such as the Core, which you mention, or the least-core), or the Nucleolous, or the Kernel, or other similar power indices such as the Banzhaf index? Are you using some of the axiomatic foundations to the Shapley value? If so, then where?

All in all, I really love the topic of the paper, but the execution could be improved (more domains for empirical evaluation, tighter theoretic bounds versus baselines). And the writing should focus on the intuitions before jumping to the technical definitions

---

> ### Author Response · Authors · 2022-08-01
> **Response to Reviewer XnpC**
>
> 1. > But why use the Shapley value rather than other solution concepts (such as the Core, which you mention, or the least-core), or the Nucleolous, or the Kernel, or other similar power indices such as the Banzhaf index? Are you using some of the axiomatic foundations to the Shapley value? If so, then where?
>
>     **Response:** We appreciate the question. First, the objective of this paper is to cope with the global reward game that is shown equivalent to the Markov convex game with the grand coalition. This leads to that we do not need to consider the coalition structure (that appears in the core or least-core). Henceforth, we just need to take into account the payoff distribution scheme (e.g. the Nucleolous, Kernel, and Banzhaf index). Among these payoff distribution schemes, we choose Shapley value since it is the only solution concept that satisfies dummy, efficiency and symmetry (i.e. the three axiomatic foundations). All these properties are important components to define fariness and validate the payoff distributions especially in real-world applications. Besides, the efficiency is an important condition to derive the Shapley-Bellman Optimality Equation (see line 135).
>
> 2. > What happens in domains which are not completely cooperative (team reward), such as social dillemas or mixed motive games. Does the algorithm still runs? Does it fail?
>
>     **Response:** This is an interesting open question and we will try our best to answer. The social dilemma is actually a motivation of the cooperative game theory [1]. In details, it happens mainly due to the lack of communications or contracts so as to fail in binding agreements. In other words, a cooperative game is to complement the failure of making binding agreements. The formal definition is as follows:
>
>     *A game is cooperative if the agents can make binding agreements about the distribution of payoffs or the choice of strategies, even if these agreements are not specified or implied by the rules of the game.*
>
>     In our paper, defining the solution concept of optimal Markov Shapley value enables binding agreements. If a game with social delimmas (e.g. Prisoners Delimma) is rewritten in the form of cooperative games (with defining coalition values by the sum of agents' payoffs where cooperation is seen as a coalition of multiple agents), then SHAQ can deal with the social dilemmas. Nevertheless, the success may hugely depend on the transformation of the game, since SHAQ was originally designed for fully cooperative games with the team reward, which violates in most games with social dilemmas.
>
> 3. > What happens when agents are trained using a mixture of algorithms (SHAQ agent with other approaches - are they compatible).
>
>     **Response:** Thanks for the interesting question. Let's consider a simple scenario where an agent is trained with SHAQ (i.e., other agents are with no Q-value update from Eq. 9) and other agents are trained with other algorithms (e.g. independent learning). In our expectation, the cooperation would collapse, since other agents would not have any incentives to form coalitions (i.e. training regardless of other agents in a coalition) in order to cooperate and Eq. 1 would not hold, which leads to the failure of learning cooperation.
>
> 4. > Can you replace the Shapley value with other cooperative solution concepts (the Banzhaf index seems to be the closest, basically your equation just with different weights for the subteams) - does the whole method fail?
>
>     **Response:** We appreciate the insightful question. Since the Banzhaf index does not satisfy the property of efficiency (that is an important condition to derive the Shapley-Bellman optimality equation in line 135), it cannot be directly substituted for Shapley value in our theoretical framework, which also verifies the compatibility of Shapley-Bellman optimality equation with Shapley value.
>
> 5. > And the writing should focus on the intuitions before jumping to the technical definitions.
>
>     **Response:** We will incoporate more intuitions partly discussed above to the revised paper.
>
>
> ### Reference
> [1] Chalkiadakis, G., Elkind, E., & Wooldridge, M. (2011). Computational aspects of cooperative game theory. Synthesis Lectures on Artificial Intelligence and Machine Learning, 5(6), 1-168.

---

> > ### Comment · Reviewer_XnpC · 2022-08-07
> > **Thanks for the comments**
> >
> > Regarding Shapley, the  Banzhaf also satisfies Dummy and symmetry, and you can normalize it to also get efficiency. Shapley also exhibits linearity over games, but not sure why this would be important here. A discussion is warranted, I think.

---

> > > ### Author Response · Authors · 2022-08-07
> > > **Further Response to Reviewer XnpC**
> > >
> > > Dear Reviewer XnpC,
> > >
> > > Thanks for your comments. We agree that the normalized Banzhaf also satisfies Dummy, Symmetry and Efficiency. If only regarding the contents discussed in this work, either index is compatible here. However, the Additivity that Shapley value possesses but the normalized Banzhaf does not, could influence the extension of the work. We will add a paragraph for discussing this in camera-ready version, thanks to the extra page permitted.

---

> ### Author Response · Authors · 2022-08-10
> **Respect to Reviewer XnpC**
>
> Dear Reviewer XnpC,
>
>
> Your knowledge on the cooperative game, such as social dilemma that studies the condition of cooperation and the cooperative game theory (seen from your comments), actually has astonished the authors.
>
> The authors would appreciate the insightful and helpful discussion with you. It is really grateful to meet you online.
>
> Respect,
>
> The Authors

---

### Official Review · Reviewer_Mapg · 2022-07-12

**Rating:** 7
**Confidence:** 3
**Soundness:** 3 good
**Presentation:** 3 good
**Contribution:** 3 good

**Summary:**

The paper presents a new framework and corresponding algorithm to solve value factorization in global reward games. Specifically, it derives the Shapley-Bellman optimality equation from evaluating the optimal Markov Shapley value and proposes the Shapley-Bellman operator to solve it, which is also proved in the paper. Furthermore, Shapley Q-learning is presented to implement the theoretical framework for predator-prey and SMAC environments.

Contributions: The paper proposes a new theoretical cooperative game framework and Shapley Q-learning algorithm for solving global reward games. Moreover, the authors give proof for the theoretical framework and evaluate SHAQ on Predator-Prey and StarCraft tasks, which shows good performance and interpretability.

**Questions:**

Authors assume games (Predator-Prey and StarCraft) are satisfied those conditions of MCG in Eq. 1 (line 78), does it always satisfy?

**Limitations:**

The assumption in line 78 for Markov convex games looks too strong, is it possible to extend the same results to general cooperative games?

**Strengths And Weaknesses:**

Strength:
1.	well written, easy to follow
2.	novel cooperative game framework for global reward game justified both theoretically and empirically
3.	well literature review on relevant fields
4.	proof details and codes provided

Weakness:
1.	Figures 1,2,3 are too small to read easily
2.	Improvements seem not significant compared to SOTAs

---

> ### Author Response · Authors · 2022-08-01
> **Response to Reviewer Mapg**
>
> 1. > Authors assume games (Predator-Prey and StarCraft) are satisfied those conditions of MCG in Eq. 1 (line 78), does it always satisfy?
>
>     **Response:** Thanks for the insightful question. Eq. 1 is a condition that describes the incentives of agents to cooperate (i.e. to form a larger coalition). If the agents in the two games are designed to cooperate, then Eq. 1 always holds. Since cooperative behaviours lead to a higher team reward, Eq. 1 is a reasonable assumption to extend the global reward game to make the credit assignment compatible and valid in it.
>
> 2. > The assumption in line 78 for Markov convex games looks too strong, is it possible to extend the same results to general cooperative games?
>
>     **Response:** Thanks for the interesting question. The answer is yes. We have discussed it in the *Limitation and Future Work*. More specifically, if we drop Eq. 1, the theoretical framework could be further extended to a more general cooperative game. The more general solution concept (e.g. core, epsilon core and etc.), however, is needed, and consequentially both the coalition structure and payoff distribution scheme are needed, than only the payoff distribution scheme in this paper.
>
> 3. > Figures 1,2,3 are too small to read easily.
>
>     **Response:** Thanks for the reviewer's notice. We will enlarge the figure size in the final version of paper.
>
> 4. > Improvements seem not significant compared to SOTAs.
>
>     **Response:** We agree that SHAQ does not improve a lot compared with SOTAs. However, we would argue that the major benefits of SHAQ is that it can provide some kind of interpretation of agents' decisions with theoretical guarantees while not weakening the performance. In the future work, we would investigate the approach to further improve the performance.

---

### Official Review · Reviewer_Bq8r · 2022-07-15

**Rating:** 6
**Confidence:** 3
**Soundness:** 3 good
**Presentation:** 3 good
**Contribution:** 3 good

**Summary:**

This paper presents a theoretical framework that studies Shapley value in the context of Markov Games as a useful technique for value factorization and credit assignment in agents coalitions. Leveraging this framework, the authors proposed Shapley Q-Learning (SHAQ) derived from a novel definition of a Shapley-Bellman Operator. The proposed algorithm is compared with a suite of existing algorithms (COMA, VDN, QMIX) in predator-prey and the StarCraft MA Challenge, contrasting competitive results while showing interesting properties of interpretability.

**Questions:**

1. Would the authors be able to reproduce the interpretability results offer for SMAC on the simple predator-pray setting?

**Limitations:**

The authors addressed the limitation of the work, including the assumptions and restrictions imposed in the scenarios considered.

**Strengths And Weaknesses:**

**Strengths**
1. The paper is well-written and properly motivated. The work is well-placed among the existing and vast literature in Multiagent Reinforcement Learning (MARL).

2. The combination of Shapley's theory with Q-Learning seems a novel contribution in the interesting and always challenging setting of MARL.

**Weaknesses**
1. The experimental section would benefit from a discussion on the interpretability of SHAQ in the predator-prey setting, which seems to be missing in the current manuscript.

---

> ### Author Response · Authors · 2022-08-01
> **Response to Reviewer Bq8r**
>
> 1. > Would the authors be able to reproduce the interpretability results offer for SMAC on the simple predator-pray setting?
>
>     **Response:** We appreciate the reviewer's advice. We have added extra demonstrations to show the interpretability of SHAQ on Predator-Prey in Appendix C.5 and the discussions are written in blue. We wish your concerns would be addressed.

---

### Author Response · Authors · 2022-08-01
**To Area Chair and All Reviewers**

Dear AC and Reviewers,

Thank you for your so insightful and helpful comments on our paper. We try our best to deal with the questions and concerns. We wish our answers will address your questions and concerns.

As Reviewer Bq8r suggested, we add additional demonstrations on Predator-Prey to verify the interpretability of SHAQ.

About writing suggestions from Reviewer Mapg and Reviewer XnpC, we will do these in the final revision of the paper.

Thanks again,

The Authors

---

### Meta-Review · Area_Chair_3fRn · 2022-08-27

**Recommendation:** Accept
**Confidence:** Certain

**Metareview:**

Reviewers appreciate that the paper is making an insightful contribution to the important field of cooperative MARL and its connection with cooperative game theory. The paper is clear and mostly well motivated, and the theoretical analysis and empirical evaluation are sufficient.

**Award:**

No

---

### Decision · Program_Chairs · 2022-09-14

Accept